# Cryo-correlative light and electron tomography of dopaminergic axonal varicosities reveals non-synaptic modulation of cortico-striatal synapses

Paul Lapios [1], Robin Anger [2], Vincent Paget-Blanc [1], Esther Marza [2], Vladan Lučić [3], Rémi Fronzes [2] ✉, Etienne Herzog [1] ✉ & David Perrais [1] ✉

Dopamine is an essential neuromodulator in the brain, involved in reward and motor control. Dopaminergic (DA) neurons project to most brain areas, with particularly dense innervation in the striatum. DA varicosities bind to target striatal synapses, forming dopamine hub synapses (DHS). However, the basic features of dopamine release sites are still largely unknown. Here we studied the ultrastructure of fluorescent DA and glutamatergic (GLU) synaptosomes isolated from the striatum of adult mice with cryo-correlative light and electron microscopy and cryo-electron tomography. We observed that DA synaptosomes display ~10 times fewer vesicles than GLU ones. DA vesicles are bigger and less round. The nanoscale organization of vesicles indicates that most GLU synaptosomes have tethered and primed vesicles, indicative of a readily releasable pool, while only 37% of DA synaptosomes have tethered vesicles, which appear not to be primed. In addition, GLU terminals contacted by DA terminals in DHS have more primed vesicles than others. While DA varicosities do not form genuine synapses, their adhesion to cortico-striatal synapses may convey a local regulation of synaptic release properties.

Neuromodulation adjusts network activity and has major impacts on behavior. Among neuromodulators, dopamine acts in the basal ganglia network to encode reward prediction and participate in the initiation of movement. Dopaminergic (DA) projections to the basal ganglia originate from two midbrain nuclei, the substantia nigra and the ventral tegmental area. These projections densely innervate the striatum to regulate the activity of spiny projection neurons (SPNs), which are central to functions like motor control, reward prediction and motivation[1]. Yet, at the ultrastructural level, the organization of DA transmission is not clear. This situation is in stark contrast with the detailed characterization of neurotransmission machineries at forebrain glutamatergic (GLU) synapses.

At GLU terminals, synaptic vesicles (SVs) are organized in a cluster polarized towards a portion of the plasma membrane called the active zone (AZ), which faces the post-synaptic density (PSD). The AZ, in which SVs are thought to fuse with the plasma membrane to release neurotransmitter, contains specific proteins such as RIM1/2, bassoon and ELKS[2]. Prior to fusion at the AZ, proximal SVs follow a series of steps, which can be observed with cryo-electron tomography (cryo-ET), from initial tethering, where SVs are tethered to the plasma membrane by one filament of ~10–25 nm[3,4], to the formation of multiple short tethers, which brings SVs closer than 5 nm to the plasma membrane[5]. These steps strongly depend on the AZ proteins RIM1 and Munc13a, which are known to control fast SV exocytosis[6]. Thus, the

[1]Univ. Bordeaux, CNRS, Interdisciplinary Institute for Neuroscience, IINS, Bordeaux, France. [2]Univ. Bordeaux, CNRS, European Institute for Chemistry and Biology, IECB, Pessac, France. [3]Department of Molecular Structural Biology, Max Planck Institute of Biochemistry, Martinsried, Germany. ✉e-mail: remi.fronzes@u-bordeaux.fr; etienne.herzog@u-bordeaux.fr; david.perrais@u-bordeaux.fr

presence of tethered and primed vesicles may constitute a hallmark of readily releasable vesicles in axons.

Dopamine is exocytosed from vesicles within milliseconds after stimulation[7,8]. This process relies on the calcium sensor Synaptotagmin-1[9,10]. Dopamine release exhibits significant paired pulse depression partially regulated by Synaptotagmin 7[11]. Moreover, DA axons contain AZ proteins, RIM1, Munc13 and ELKS, which are important for fast dopamine release[12,13]. However, only ~30% of DA varicosities contain such assemblies[12,14]. Intriguingly, uptake of fluorescent dopamine analog in ex vivo striatal slices shows that only ~25% of all DA varicosities release it after electrical stimulation[15], which suggests functional heterogeneity of these terminals. Furthermore, serial electron microscopy studies of DA axon terminals in mouse striatum revealed heterogeneous vesicular content, with terminals containing varying combinations of small and large vesicles, while some varicosities appeared to lack vesicles altogether[16]. Overall, DA terminals exhibit a non-stereotypical vesicular organization, which may explain the functional diversity of release. However, observations of the precise spatial arrangement of DA vesicles prior to fusion have been hampered by chemical fixation, staining procedures and difficulties in identifying DA axons. These methodologies affect the preservation of cellular morphology and preclude interpretation of molecular details[17].

Despite significant advancements in the field, the relationship between DA release sites and target cells remains poorly documented. Upon release, dopamine binds only to G-protein coupled receptors of the D1 group (D1/5 R) or D2 group (D2–4R), whose signals respectively increase or decrease the excitability of target cells. On presynaptic terminals, they influence SV release probability, while at the post-synapse, they act on ion channels and glutamate receptors (reviewed in ref. [18]). In particular, neurons projecting from motor and prefrontal cortical regions form GLU cortico-striatal (CS) synapses responsible for the activation of SPNs. Interestingly, the stimulation of dopamine release in acute slices attenuates the release kinetics from a subset but not all GLU terminals[19], showing that dopamine influences the activity of CS synapses. Like other neuromodulators, dopamine released from a single release site could influence large neuronal assemblies in a so-called volume transmission[20]. However, the precise ultrastructure through which DA terminals interact with their synaptic targets is unclear. DA synapses showing classical pre and post-synaptic features represent a minority[16,21–23]. Nevertheless, DA boutons are often found in close apposition with either pre or post-synaptic elements (PSEs) of CS synapses[24]. This proximity is functionally relevant because DA release generates micrometer-wide hotspots of dopamine as estimated using carbon nanotube or genetically encoded dopamine sensors[7,8,25]. Moreover, the synchronization of DA phasic release with local glutamate uncaging induces structural plasticity at spines[26].

Isolating DA synaptosomes from striatal tissue and sorting them by fluorescence-activated synaptosome sorting (FASS) revealed that most DA terminals are in close contact with other terminals, such as GLU presynapses[27]. These interactions are conserved even though tissue homogenization and droplet-based fluorescence-activated sorting exposed structures to significant mechanical shearing forces. We termed these multipartite DA-containing synapses "dopamine hub synapses" (DHS). Among them, 25% were formed with CS synapses marked by the vesicular glutamate transporter VGLUT1[24,27,28]. Importantly, CS-DHS contain an increased signal for the presynaptic proteins VGLUT1 and bassoon compared to other CS synapses[27]. Collectively, these findings highlight the importance of local DA signaling for the regulation of CS synapses. However, there is still a major lack of ultrastructural observations in close-to-native conditions of DA terminals associated with their target synapses.

Cryo-correlative light and electron microscopy (cryo-CLEM) and cryo-electron tomography (cryo-ET) enable identification of vitrified fluorescently labeled terminals and close-to-native 3D observations of cellular ultrastructure and protein complexes at a nanometer scale[29–31]. This allows determination of the vesicular organization in different synaptic types and characterization of protein complexes[32]. Synaptosomes are a suitable model for cryo-EM because they can be vitrified by plunge freezing, imaged in transmission EM without thinning, and because they preserve the association between terminals and important pre- and post-synaptic function[17,31,33]. Here, we applied cryo-CLEM and cryo-ET to fluorescently labeled synaptosomes in order to determine the ultrastructural features of DA terminals, CS synapses and CS-DHS. We reveal the spatial organization of DA vesicles, as well as the structural association with GLU synapses in DHS. Importantly, we observed tethered SVs in DA terminals and quantified differences in SV organization of CS that are correlated with the association with DHS.

## Results
### Identification and observation of GLU and DA synaptosomes by cryo-CLEM and cryo-ET

Using cryo-CLEM combined with cryo-ET, we assessed the ultrastructure of identified GLU and DA terminals extracted from the striatum of adult mice (Fig. 1). We prepared synaptosomes from the striata of 12–18-week-old mice as previously described[27] and detailed in Methods and Fig. 1A, B from three different mouse lines. For GLU elements, we took advantage of the knock-in (KI) mouse line in which the VGLUT1 open reading frame is tagged with the sequence of the fluorescent protein Venus[34]. In the striatum, VGLUT1[venus] specifically labels CS terminals[24,28]. For DA synaptosomes, we injected an adeno-associated viral vector carrying sequences for Cre-dependent mNeonGreen expression (AAV1 pCAG-Flex-mNeonGreen) in the midbrain of dopamine transporter promoter (DAT)-Cre transgenic mice, which specifically labels DA neurons[27]. To identify CS-DHS, we used dual tagging of GLU (green) and DA (red) synaptosomes by crossing VGLUT1[venus] mice with DAT-Cre[35] and the reporter line Ai14tdTomato[36]. Striatal synaptosomes were incubated at 37 °C for 15 minutes before use. They are capable of depolarization-evoked exocytosis and recycling, as shown by uptake and release of the membrane dye FM4-64 (Fig. S1), in accordance with previous work[37,38]. We mixed this suspension with electron-dense fluorescent fiducial beads for alignment and gold beads for tomogram reconstruction. We applied the suspension on an EM grid, plunge-froze it into liquid ethane (Fig. 1C) and kept it in liquid nitrogen for further use.

We first observed the grids with a cryo-fluorescence microscope. We selected isolated fluorescent spots with nearby fiducial beads for alignment (Fig. 1D). In mice with both fluorescent markers for GLU and DA synaptosomes, we selected either an individual isolated spot or pairs of spots where the two colors were separated by <1 μm, corresponding to putative DHS. We then observed the same grids with a cryo-transmission electron microscope and located regions of interest based on bead patterns (Fig. 1E). We used the images of neighboring independent fiducial beads to measure the pointing precision between cryo-fluorescence and cryo-electron microscopy (Fig. S2; 117 ± 82 nm, $n = 95$). This pointing precision is smaller than the radii of GLU and DA synaptosomes (see below). Therefore, we can identify unambiguously the terminals of interest in cryo-EM, either GLU or DA. Finally, we acquired a tilt series of selected synaptosomes and reconstructed tomograms with the weighted back-projection method. We obtained tomograms from 20 preparations (Table S1): 4 from VGLUT1-Venus mice (41 GLU synaptosomes), 6 from DAT-Cre+AAV-mNeonGreen mice (59 DA synaptosomes) and 10 from VGLUT1-Venus*DAT-Cre-Ai14tdTomato (62 GLU and 51 DA synaptosomes, among which 32 form DHSs). We thus have a dataset of tomograms of 103 GLU synaptosomes and 110 DA synaptosomes with 32 CS-DHSs (Fig. 1F). We generated 3D models of these synaptosomes, in which we segmented the plasma membrane, internal membranes and, when applicable, identified adhering structures such as a PSE, as illustrated in Fig. 1G.

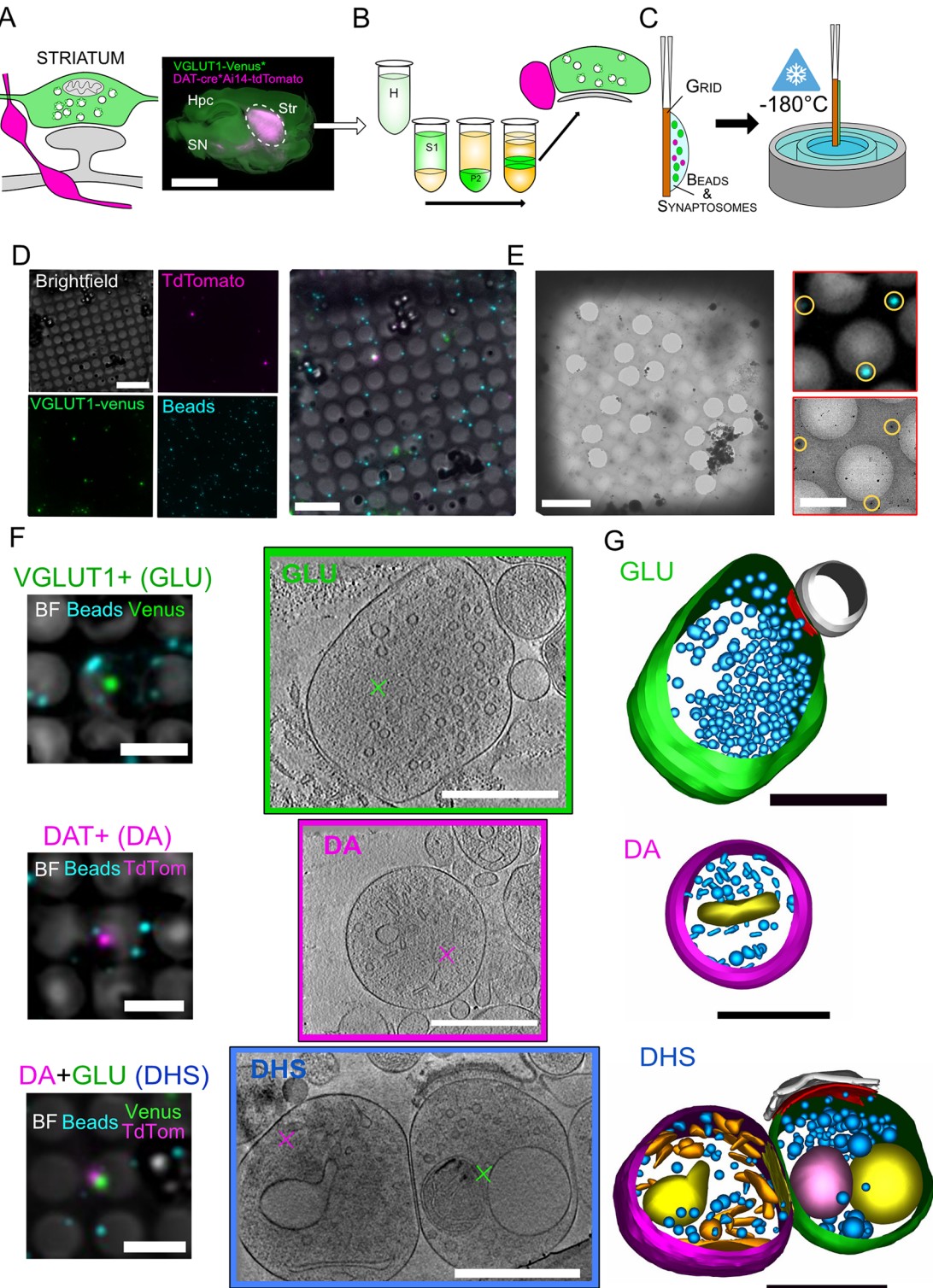

## GLU synaptosomes are larger and contain more SVs than DA synaptosomes

GLU synaptosomes have a mean size, measured by their maximal extension, of 823 nm. It corresponds to a visible volume of $0.0915\ \mu m^3$ (Fig. 2A, C, D). All GLU synaptosomes contain small, round SVs (Fig. 2A). The number of vesicles per synaptosome ranges from 13 to 872, 191 on average (Fig. 2E). Therefore, synaptosomes have, on average 1956 vesicles/$\mu m^3$ (Fig. 2F). In addition, 21/103 synaptosomes contain a mitochondrion (Figs. 1F and 2A). The synaptosomes with a mitochondrion are significantly larger than the ones without (max

extension $1045 \pm 205$ nm vs $785 \pm 250$ nm, $p < 0.0001$). Some synaptosomes contain organelles such as round vacuoles, multivesicular bodies, and clathrin-coated vesicles, as well as other intracellular features, such as filaments (Fig. S3A–F). The majority of GLU synaptosomes (64/103, 62%) are connected to a PSE. PSEs also occasionally contain intracellular organelles (Fig. S3G–I). A gallery of representative tomographic slices and 3D models of GLU synaptosomes is shown in Fig. S4.

DA synaptosomes (example Fig. 2B) have a mean size of 575 nm, which corresponds to a visible volume of $0.0410\ \mu m^3$, significantly

**Fig. 1 | Cryo-CLEM and cryo-ET of cortico-striatal and dopaminergic synaptosomes. A** Left, schematic of the expected organization between DA varicosities (magenta) and GLU synapses (green), forming DHSs. Right, fluorescence of a sagittal slice of VGLUT1^venus*DAT-Cre*Ai14tdTomato mouse. CS synapses and DA projections generate intense fluorescence signals in the striatum (Str, dotted line). Hpc Hippocampus, SN Substancia Nigra; Scale bar: 3 mm. **B** Diagram of synaptosome preparation from fresh striatum obtained after differential centrifugations. The fractions (H, S1, P2, B, see Methods) are increasingly enriched in synaptosomes. **C** A suspension of synaptosomes mixed with fluorescent and gold fiducial beads is layered on the grid, blotted and plunge-frozen in liquid ethane. **D** Example of a cryo-fluorescence microscopy image of a single grid square from a VGLUT1^venus x DAT-Cre x Ai14tdTomato with four channels: brightfield, fluorescence of tdTomato, Venus and blue fiducial beads. These images were used to select synaptosomes of interest, green (VGLUT1^venus) or magenta (DAT+), next to several blue beads, for observation with electron microscopy. Scale bar 10 μm. **E** Left cryo-EM image of the same square as displayed in (**D**). Scale bar 10 μm. Right, subset of cryo-fluorescence (top) and EM (bottom) of the same portion of the grid with three fiducial blue beads visible in both modalities (yellow circles) for fine registration. Scale bar: 3 μm. Synaptosomes with clearly visible beads and good ice quality were further imaged with high-magnification tilted series. **F** Reconstructed tomograms of fluorescent synaptosomes obtained by cryo-electron tomography. Left, overlay of fluorescence and brightfield channels showing tagged synaptosomes of interest and fiducial beads. Scale bars: 5 μm. Right, representative single tomographic slice of 1.558 nm of thickness showing a clear structure outline and intracellular organelles. Magenta or green cross marks point to the registered center of the fluorescent tag (GLU or DA synaptosome), Scale bars: 500 nm. **G** 3D models from the synaptosomes displayed in (**F**) were obtained by segmentation of membranes. The display of 3D models are as follows: plasma membrane of synaptosomes in green for GLU or magenta for DA; PSE in gray; active zone (AZ) in red; Small (synaptic, <80 nm) vesicles in light blue; large endosome-like organelles (>80 nm) in yellow; elongated endoplasmic-reticulum-like structures in orange; mitochondrion in pink; multivesicular bodies in dark blue (see Fig. 2). For both GLU synaptosomes, a clear synaptic cleft was detected, with either a closed post-synaptic element (PSE) in gray (top), or an opened post-synaptic membrane (bottom). Both of them exhibit a post-synaptic density. The presynaptic GLU active zones are shown in red. DA synaptosomes originate from 16 different preparations, GLU synaptosomes from 14 preparations, and DHS from 10 preparations, see Table S1. Scale bar: 500 nm.

smaller than GLU synaptosomes (Fig. 2C, D, $p$ value < 0.0001). DA synaptosomes contain, on average 28 vesicles, but their number is quite variable: some are almost empty (27/110 synaptosomes have <5 vesicles), whereas others contain tens or even hundreds of vesicles (Figs. 2E and S5, S6A, B). Overall, the vesicle density of DA synaptosomes is ~threefold smaller (682 per μm³) than for GLU synaptosomes 1964 per μm³) (Fig. 2F). DA synaptosomes also contain other organelles, such as large endosomes/vacuoles, mitochondria, multivesicular bodies (Figs. 1F, 2B and S6C–H). DA synaptosomes containing mitochondria (9/110) are significantly larger than the ones who do not (max extension $838 \pm 224$ nm vs $555 \pm 176$ nm, $p < 0.0001$). Table S2 reports the number of DA synaptosomes with these organelles.

While SVs in GLU synaptosomes are spherical, small and uniform in size (outer diameter $40.4 \pm 7.0$ nm, $n = 20087$, with only 2.2% vesicle larger than 60 nm), vesicles in DA synaptosomes are significantly larger ($45.1 \pm 9.7$ nm, $n = 3004$, $p < 0.0001$) with 8.9% of vesicles larger than 60 nm (Fig. 2H). In total, 71/110 DA synaptosomes have at least one vesicle bigger than 60 nm, with a proportion of $28.5 \pm 21.2\%$ big vesicles in these synaptosomes. Moreover, vesicles in DA synaptosomes are often elongated or pleomorphic (Fig. 2I). We used Wadell's index WI (see Methods) to quantify vesicle sphericity. Most vesicles in GLU terminals have an index close to 1 (perfect sphere). However, vesicles in DA synaptosomes are significantly less round than the ones in GLU synaptosomes (Fig. 2I, $p$ value < 0.0001). These elongated vesicles are spread across the synaptosomes (Fig. S5). Overall, there are 72/110 DA synaptosomes containing at least one elongated vesicle (WI < 0.95). In these synaptosomes, there are $29.4 \pm 27.3\%$ elongated vesicles. Notably, although small (diameter <80 nm) vesicles are clearly different between GLU and DA synapses, large endosome-like vesicles (diameter >80 nm) have similar sizes ($187 \pm 122$ nm $n = 63$ vs $179 \pm 67$ nm $n = 32$, for GLU and DA synaptosomes, respectively $p = 0.47$).

### Contact zones between GLU, DA synaptosomes and PSEs
In GLU synaptosomes, we identified a clear PSE separated from a presynaptic terminal by a clearly defined synaptic cleft in 64/103 (62%) synaptosomes. Typically, the PSE is defined by a sealed plasma membrane compartment (Figs. 1F, 3A, C and S4), but in some cases, it had an open membrane adhering to the presynaptic terminal forming a synaptic cleft (Fig. 1F, S4). The area of contact between GLU and PSE is characterized by roughly parallel membranes defining a wide cleft ($31.8 \pm 6.2$ nm, $n = 34$) with dark material around the midline (Fig. 3D, E), as observed previously[39,40]. The AZ, defined as the membrane region of GLU synaptosome in contact with the PSE (outlined in red in Fig. 3A), has an area of $0.079 \pm 0.055$ μm² ($n = 32$) (Fig. 3F). We analyzed

the electron density in close proximity to the pre- and post-synaptic membranes of GLU synaptosomes (Fig. 3J, K). This may reflect the density of proteins and lipids in the area. The density is maximal in the vicinity of the plasma membrane. At the presynapse, we measured a decrease by ~20% in <10 nm away from the plasma membrane. In the PSE, we detected, in addition to the decrease within 10 nm, a clear increase in density 10–25 nm from the PSE membrane, which corresponds to the PSD as detected with other modalities of electron microscopy in situ[41] and synaptosomes with cryo-ET[31]. In contrast, only two out of 110 DA terminals are separated by a wide cleft from a characteristic PSE containing PSD, as in GLU synaptosomes. Interestingly, one of these DA synaptosomes has the most vesicles (610), all of them small and round, which makes it indistinguishable from classical GLU synaptosomes (Fig. S6A). The other one has 39 vesicles (Fig. S6B). In these two synaptosomes, the distribution of vesicle sizes and sphericity matches distributions seen for GLU synaptosomes (dotted line in Fig. 2H, I). Another DA synaptosome has 295 vesicles (see dot in Fig. 2E) and also resembles classical GLU synaptosomes, even though it is not connected to a PSE but engaged in a CS-DHS. These DA synaptosomes could originate from neurons co-expressing VGLUT2 and which are able to release glutamate and generate AMPAR-mediated post-synaptic potentials[42,43].

Nevertheless, many DA terminals are in close apposition with other structures, which occasionally resemble a GLU terminal (Fig. S5A), or other, harder to characterize structures, which could be parts of post-synaptic spines. To get a better insight into the features of contact zones between DA and GLU terminals, we analyzed 32 tomograms from apposed GLU and DA synaptosomes, which correspond to CS-DHSs[27]. Among them, 28 show a contact between the DA terminal and the GLU presynapse (87%), and the other four show a contact between the DA terminal and the GLU post-synapse (13%). The contact site of DA and GLU presynapse is clearly distinguishable from the GLU AZ, as they are separated by a distance of $860 \pm 389$ nm along the plasma membrane. The area of contact between GLU and DA synaptosomes has a similar size to GLU AZs (Fig. 3B, F). The width of the cleft between GLU and DA terminals is $12.1 \pm 2.4$ nm, much smaller than the synaptic cleft of GLU synaptosomes (Fig. 3G, $p < 0.0001$). Along this cleft, dense material is also observed, albeit not as pronounced as for GLU/PSE synaptic clefts (Fig. 3H, I). The intracellular side of DA and GLU terminals around the adhesion site is devoid of visible densities such as PSDs (Fig. 3L, M). In two tomograms, the DA terminal is not in contact with the presynaptic GLU synaptosome but with the PSE (Fig. 3C and S7B). The contact area (0.062 and 0.066 μm²) and cleft sizes (10.9 and 15.6 nm) are in the same range as for DA/preGLU contact sites (Fig. 3F, G). Similarly, the densities around the

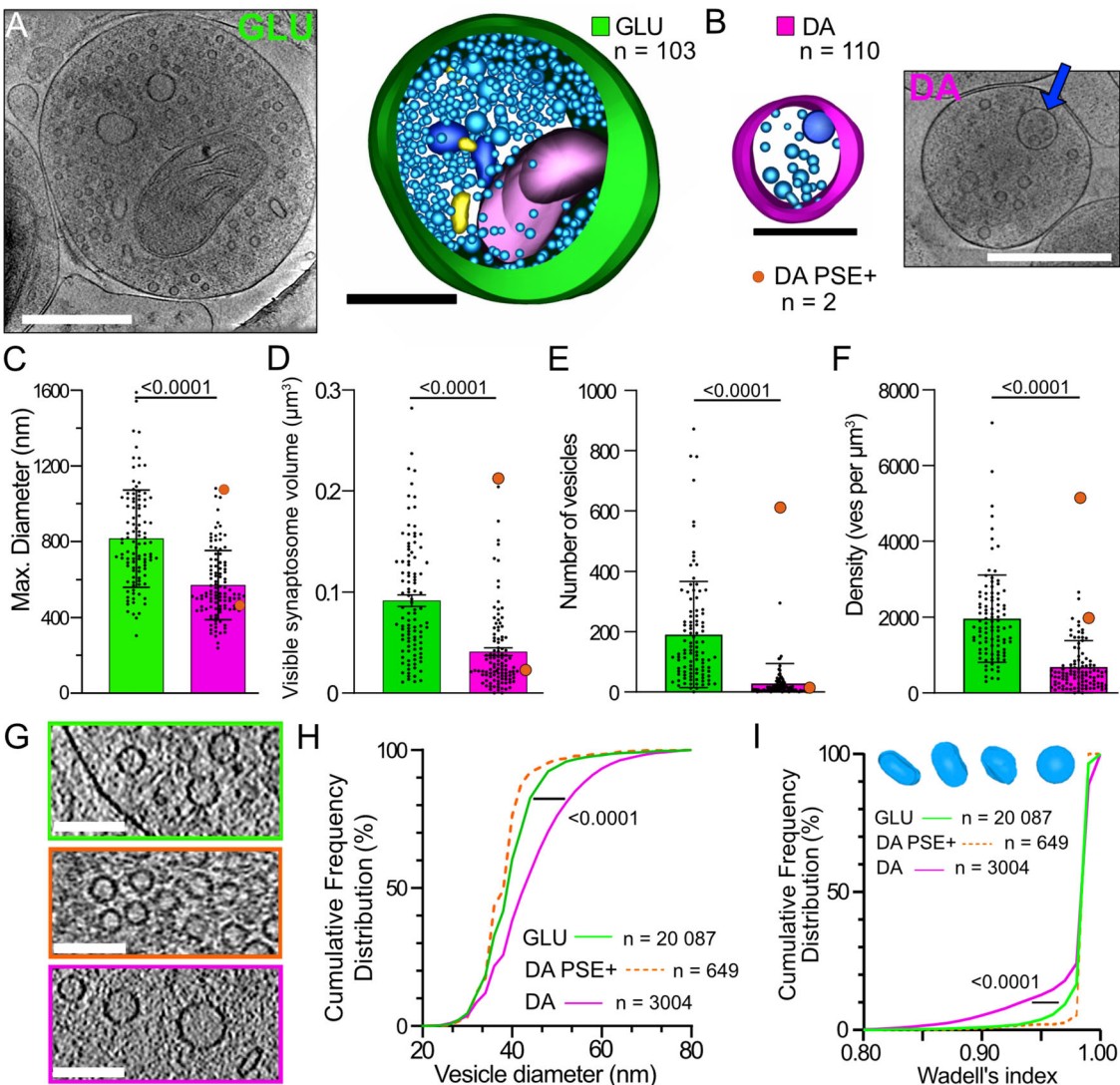

**Fig. 2 | Comparison of synaptosome size and vesicular content for GLU and DA synaptosomes. A, B** Example of a single tomogram plane of 1.558 nm thickness, and corresponding to 3D models of GLU (**A**) and DA (**B**) synaptosomes. Same color code as in Fig. 1. Scale bars 500 nm. **C–F** Distributions and maximal diameters of synaptosomes (**C**), visible volumes (**D**), number of vesicles (**E**) and vesicle densities (**F**) of 103 GLU and 110 DA synaptosomes. The distributions are all different (Two-tailed *t* test *p* < 0.0001). In the histograms of DA synaptosomes, orange dots represent the DA terminals with a PSE sharing the features of asymmetric synapses (examples in Fig. S6). Bar charts represent mean values ± SD. **G** Cropped areas in tomograms of GLU (green), DA (magenta) and DA with a PSE (orange) synaptosomes display the variety of vesicle sizes and shapes. Scale bars 100 nm. **H** Cumulative distribution of vesicle diameters in GLU (green line, *n* = 20087) and DA (magenta line, *n* = 3004) synaptosomes. The diameters are significantly different (Two-tailed Kolmogorov–Smirnov test *p* < 0.0001). The orange dotted line (*n* = 649) corresponds to vesicles in DA terminals with a PSE. **I** Cumulative distribution of vesicle sphericity (WI). GLU vesicles are significantly more spherical than DA vesicles (Two-tailed Kolmogorov–Smirnov test *p* < 0.0001). We show examples of DA vesicles with WI = 0.85, 0.89, 0.95, and 0.99 from left to right above the plot.

contact site are in the same range, without signs of a PSD (Fig. 3N, O). More examples of DHS virtual planes and models are available in Fig. S7.

## Vesicles in GLU and DA terminals are tethered to the plasma membrane and are interconnected

SVs in GLU and GABAergic synapses are divided into functional pools[44]. SVs, which fuse first with the plasma membrane upon calcium stimulation, comprise the readily releasable pool, which was defined morphologically by EM of chemically fixed, dehydrated samples as the vesicles that appear docked, that is, in direct contact with the plasma membrane[45,46]. Cryo-ET revealed that in close to native state, SVs are not docked but are tethered to the plasma membrane via one or several electron dense filaments, as observed in synaptosomes and dissociated neuronal cultures[3,4,31,32,47]. We defined proximal vesicles as

those located within 45 nm of the AZ, where we sometimes observe tethers (Fig. 4A). Tethers, as well as connectors (interconnecting SVs, Fig. 4A), were detected using an automated, template-free method by the hierarchical connectivity algorithm[48]. Among GLU proximal vesicles, 52% are tethered (Fig. 4C). These tethers have a length of 13.9 ± 7.5 nm (Fig. 4D). The number of tethers increases as vesicles are located closer to the plasma membrane (Fig. 4E). Remarkably, vesicles located <5 nm from the plasma membrane have on average 3 tethers, which corresponds to SVs that are functionally primed for fusion[3,4].

In DA synaptosomes, we found clear examples of tethers between vesicles and the plasma membrane, as well as vesicles connected by more than one tether (Fig. 4B). We detected vesicles tethered to the plasma membrane in 37% (18/49) of DA synaptosomes. The percentage of tethered vesicles among proximal vesicles (within 45 nm of the plasma membrane) in DA synaptosomes is two times lower than for

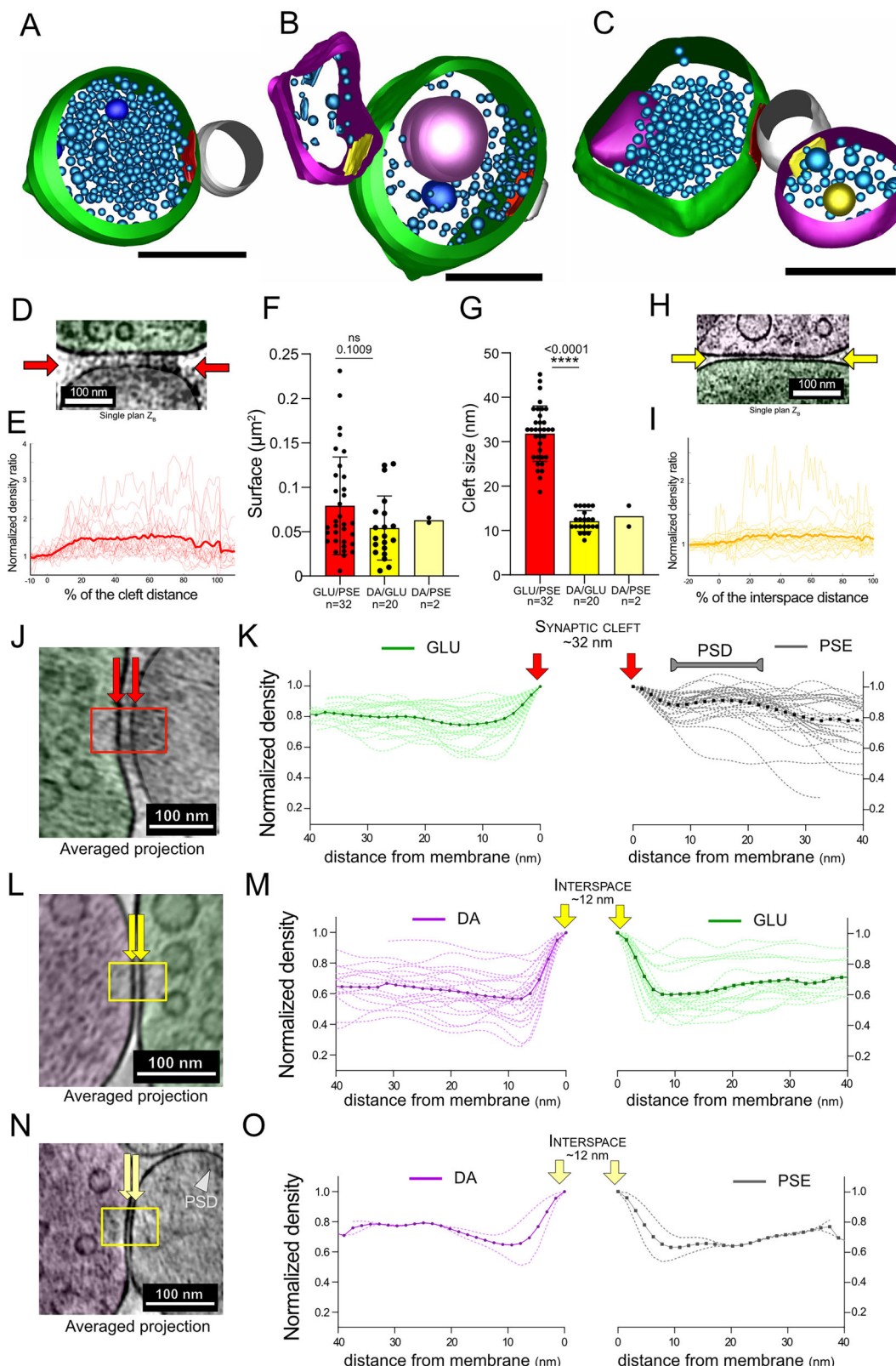

GLU synaptosomes (25%) (Fig. 4C, *p* < 0.0001). However, tether length is on average significantly larger in DA versus GLU (respectively, 22.4 ± 12.7 nm vs 13.9 ± 7.5 nm; *p* value < 0.0001) (Fig. 4D). This difference is possibly a direct consequence of the almost complete absence of tethered DA vesicles localized <5 nm from the plasma membrane (Fig. 4E). Effectively, we found only 1 out of 135 vesicles closer than 5 nm. This vesicle has two tethers. Importantly, for DA and GLU vesicles

located further away, the number of tethers per SV is not different (Fig. 4E).

Another morphological hallmark of the polarized exocytosis of SV in cortical synaptosomes is the larger fraction of volume occupied by proximal SVs towards the AZ[3,4,31]. GLU synaptosomes show a mean peak occupancy ~20 nm from the plasma membrane that is clearly visible in individual occupancy profiles (Fig. 4F, G) and is similar to

**Fig. 3 | Contact zones between GLU, DA synaptosomes and PSEs. A–C** 3D models of a GLU presynaptic element (green) connected to a PSE (gray). The active zones are red. **B** A DA element connects the GLU presynaptic element. **C** A DA element connects the PSE. The contact zones with DA are yellow. Scale bars 500 nm.
**D** Single plane showing the synaptic cleft between a GLU synaptosome (outlined in green) and a PSE (outlined in gray). The electron-dense material in the middle of the cleft is clearly visible (red arrows). **E** Pixel intensities along the length of the synaptic cleft (between the red arrows in **D**), normalized to neighboring background ($n = 20$). **F** Area of active zones of GLU synaptosomes and contact zone between DA and GLU synaptosomes or DA and PSEs (GLU/PSE: $0.0791 \pm 0.0550$ μm² vs. DA/GLU: $0.0541 \pm 0.0360$ μm²; two-tailed t test, $p$ value = 0.101) (DA/PSE: $0.0627 \pm 0.0388$ μm²). **G** Membrane to membrane distance for: GLU with PSE (synaptic cleft, $31.8 \pm 6.2$ nm), DA with GLU ($12.1 \pm 2.3$ nm, two-tailed $t$ test vs. GLU/

PSE, $p$ value < 0.0001), and DA with PSE ($13.2 \pm 3.3$ nm). **H** Single plane showing the cleft between a DA (magenta) and a GLU synaptosome (green). **I** Pixel intensities along the length of the DA/GLU contact site (between the yellow arrows in **H**), normalized to the region outside the contact site ($n = 24$). **J** Image of a contact site (synaptic cleft) between GLU and PSE. **K** Normalized intensity profiles ($n = 14$) in the cytoplasm starting 5 nm away from the plasma membrane, towards the center of GLU (green) and PSE (gray). In both cases, the averaged intensity reached a plateau -80% 40 nm away. In the PSE, the intensity clearly increases 20 nm away, which corresponds to the PSD. We observe no clear increase in cytoplasmic density on the presynaptic side of GLU. **L, M** Same as **J, K** for DA-GLU contact sites, ($n = 10$). **N, O** Same as **J, K** for DA-PSE contact sites, ($n = 2$). All bar charts represent mean values ± SD.

those observed for cortical synaptosomes[3,4,31]. In DA synaptosomes, the occupancy lacks the peak in the proximal vesicles region, but occasional enrichment close to the whole plasma membrane is observed (Fig. 4H, I). However, the absence of a morphologically identifiable DA AZ may have biased the measurement of occupancy, because it included a larger, possibly irrelevant volume in our analysis.

The percentage of inter-connected SVs is significantly higher in GLU (64%) than in DA synaptosomes (24%) (Fig. 4J, $p < 0.0001$), and the length of connectors is significantly different ($18.5 \pm 9.2$ nm vs $14.9 \pm 11.1$ nm for DA and GLU, respectively; $p < 0.0001$; Fig. 4K). Finally, among proximal vesicles, 17% of GLU SVs are both tethered and connected vs 8.1% in DA synaptosomes. In the latter, the majority of vesicles are neither tethered nor connected (Fig. 4L).

### Tethered SVs highlight a subpopulation of DA terminals
We compared DA synaptosomes that contain (T+) to ones that do not contain (T−) tethered vesicles (Fig. 5A). We excluded from the analysis the two DA synaptosomes with a PSE (see Fig. S6A, B). The T + DA synaptosomes contain significantly (three times) more SVs (Fig. 5B, $p < 0.0001$), and are significantly larger than the T− synaptosomes (Fig. 5C, $p$ value = 0.0013). Therefore, T + DA synaptosomes have a larger vesicle density than T− synaptosomes (Fig. 5D, $p$ value = 0.0005). Moreover, in T + DA synaptosomes, vesicles are located closer to the plasma membrane than in T− DA synaptosomes (Fig. 5E). We also compared the diameter and sphericity of the 43 tethered vesicles with the whole population and found that tethered vesicles are larger (Fig. 5F, $p = 0.0016$) and rounder (Fig. 5G, $p < 0.0001$).

In T + DA synaptosomes, we do not observe a clear AZ, which corresponds in GLU synapses to the location facing the synaptic cleft where tethered SVs concentrate and undergo exocytosis upon stimulation. We looked for a putative DA AZ, which is a region of the plasma membrane where vesicle fusion may occur preferentially, in two ways. First, the AZ could be located at or right next to the contact zone with GLU terminals. We found only one tethered vesicle at the DA/GLU contact site, while the other 44 tethered vesicles are located at distances ranging from 250 to 750 nm. The average distance of these tethered vesicles from the contact site is $479.8 \pm 292.2$ nm (Fig. 5H). Second, we expect that an AZ would concentrate tethered vesicles. Therefore, we analyzed DA synaptosomes with multiple tethered vesicles. T+ synaptosomes contain $2.5 \pm 1.2$ tethered vesicles, and a maximum of 5 (Fig. 5J). Among the 18 T + DA synaptosomes, nine had multiple tethered vesicles, for a total of 36 tethered vesicles (Fig. 5I, J). The shortest arc distance (i.e., staying on the plasma membrane) between these vesicles is $216 \pm 193$ nm (Fig. 5K). This is smaller than the average distance between random points on the synaptosome plasma membrane (estimated at $567 \pm 240$ nm, see Methods for derivation). This suggests that tethered DA vesicles may gather in a preferential zone of the plasma membrane, a putative AZ. On the other hand, in GLU synaptosomes, the arc distance between tethered SVs is $100 \pm 12$ nm, significantly smaller than in DA terminals (Mann–Whitney

test, $p < 0.0001$). Therefore, the factors clustering tethered SVs in GLU AZ are stronger than the ones in DA terminals, if any.

### Correlation of the DHS connection with alterations of CS terminals
We wondered whether the adhesion of a DA terminal to a GLU synapse affects the ultrastructure of the GLU presynapse. We compared GLU synaptosomes that were not part of DHS (GLU DA− synapses, Fig. 6A, $n = 30$) with GLU synaptosomes engaged in a DHS (GLU DA+ synapses or DHSs, Fig. 6B, $n = 32$), both from the dual color model. The total number of vesicles is not different between GLU DA − and GLU DA+ terminals, on average 205.7 and 222.9 vesicles (Fig. 6C; $p$ value = 0.864), as well as, the density of vesicles, 1893 per μm³ and 1943 per μm³, respectively (Fig. 6D; $p$ value = 0.577). Similarly, the AZ areas are not different (Fig. 6E). Together, these argue that the morphology of terminals and overall distribution of SVs is the same.

Nevertheless, the fraction of the synaptosome volume occupied by proximal SVs is significantly higher in GLU DA+ than in GLU DA− synaptosomes, and shows a peak that signifies a higher concentration of proximal SVs, while the occupancy peak is absent in GLU DA-terminals (Fig. 6F, G). A precise characterization of SV location and tether length shows that the results for GLU DA+ terminals are similar to those of unperturbed hippocampal GLU synapses reported previously[3], while GLU DA− terminals have a proportion two times lower of SVs located 5–10 nm to the AZ membrane (Fig. S10).

Additionally, proximal vesicles in GLU DA+ synaptosomes are significantly more connected to other vesicles than in GLU DA− synaptosomes (46.7% vs. 21.6%; $p$ value = 0.0183) (Fig. 6J). The fractions of proximal vesicles that are tethered are not statistically different between GLU DA+ (61%) and GLU DA− (47%) (Fig. 6H, $p$ value = 0.204), and the number of tethers among them is unchanged (Fig. 6I). However, the fraction of proximal SVs that are both tethered and connected is five times higher in GLU DA+ terminals (26.7% vs. 5.4%) (Fig. 6K, L). Together, our results suggest that connectors are primarily responsible for the observed difference in the proximal SV distribution.

Finally, we tested the possibility that the participation of a DA synaptosome in a CS-DHS could affect the distribution of its vesicles. We found no significant differences in their vesicle number, spatial organization and tethering (Fig. S9). This suggests that the adhesion to a GLU terminal does not affect the propensity of DA synaptosomes to contain tethered vesicles.

### Discussion
Here, we report the ultrastructure of fully hydrated, close-to-native DA terminals from adult mouse striatum at a single nanometer resolution, as well as of DHS, which comprise DA terminals in contact with GLU synapses and were characterized previously with immunofluorescence[27].

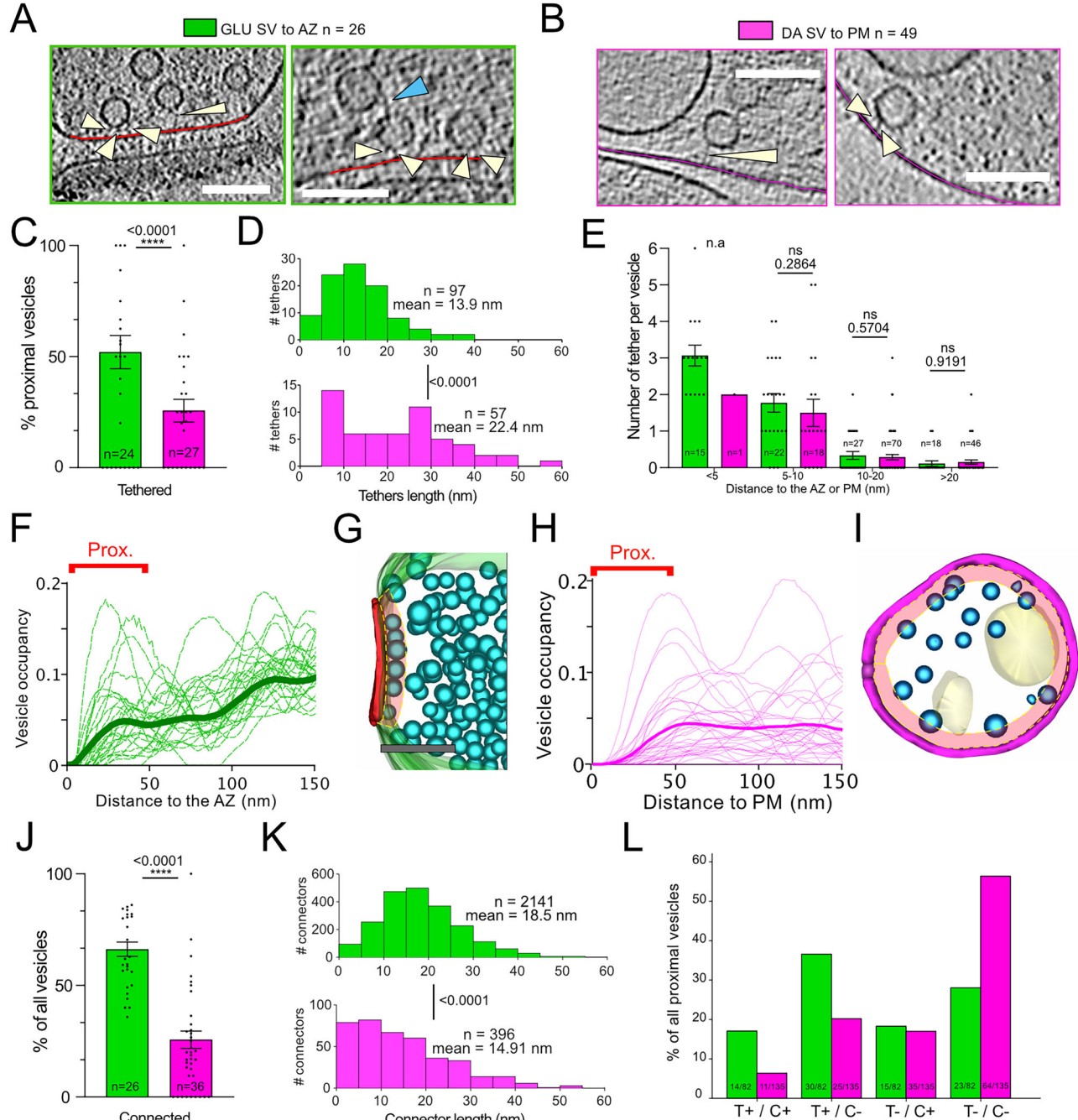

**Fig. 4 | Vesicles with tethers and connectors in GLU and DA synaptosomes.**
**A** Tomogram plane showing vesicles tethered to the plasma membrane in GLU synaptosomes (white arrowheads). Some vesicles are connected via multiple tethers. Connectors between vesicles are also visible (light blue arrowhead). Scale bar 100 nm. **B** Same as (**A**) for DA synaptosomes. **C** Percentage of proximal vesicles (located <45 nm) tethered in GLU and DA synapses (respectively, 52% and 25%, one-sided chi² test; $p$ value < 0.0001, data points for each synaptosome are shown). **D** Length of tethers connecting vesicles to the plasma membrane in GLU and DA synapses (Two-tailed Mann–Whitney; $p$ value < 0.001). **E** Average number of tethers for vesicles at various distances from the plasma membrane (Two-tailed $t$ tests; $p$ values = 0.934; 0.777; 0.770). **F** Fraction of volume occupied by vesicles vs distance

to the AZ for GLU synapses. The mean distribution is shown as a thick line. **G** 3D model of a GLU synaptosome around the active zone showing the concentration of proximal vesicles that are tethered. **H** Fraction of volume occupied by vesicles vs distance to the plasma membrane for DA synapses. The mean distribution is shown as a thick line. **I** 3D model of a DA synaptosome with the zone of proximal vesicles. **J** Percentage of all vesicles connected to another in GLU and DA synaptosomes (respectively, 64% and 24%, one-sided chi² test; $p$ value < 0.0001, data points correspond to the percentage for each synaptosome). **K** Average length of connectors (two-tailed $t$ test; $p$ value < 0.0001). **L** Proportion of proximal vesicles that are tethered and/or connected in GLU and DA synaptosomes. Bar charts represent mean values ± SEM (**C**, **E**, **J**).

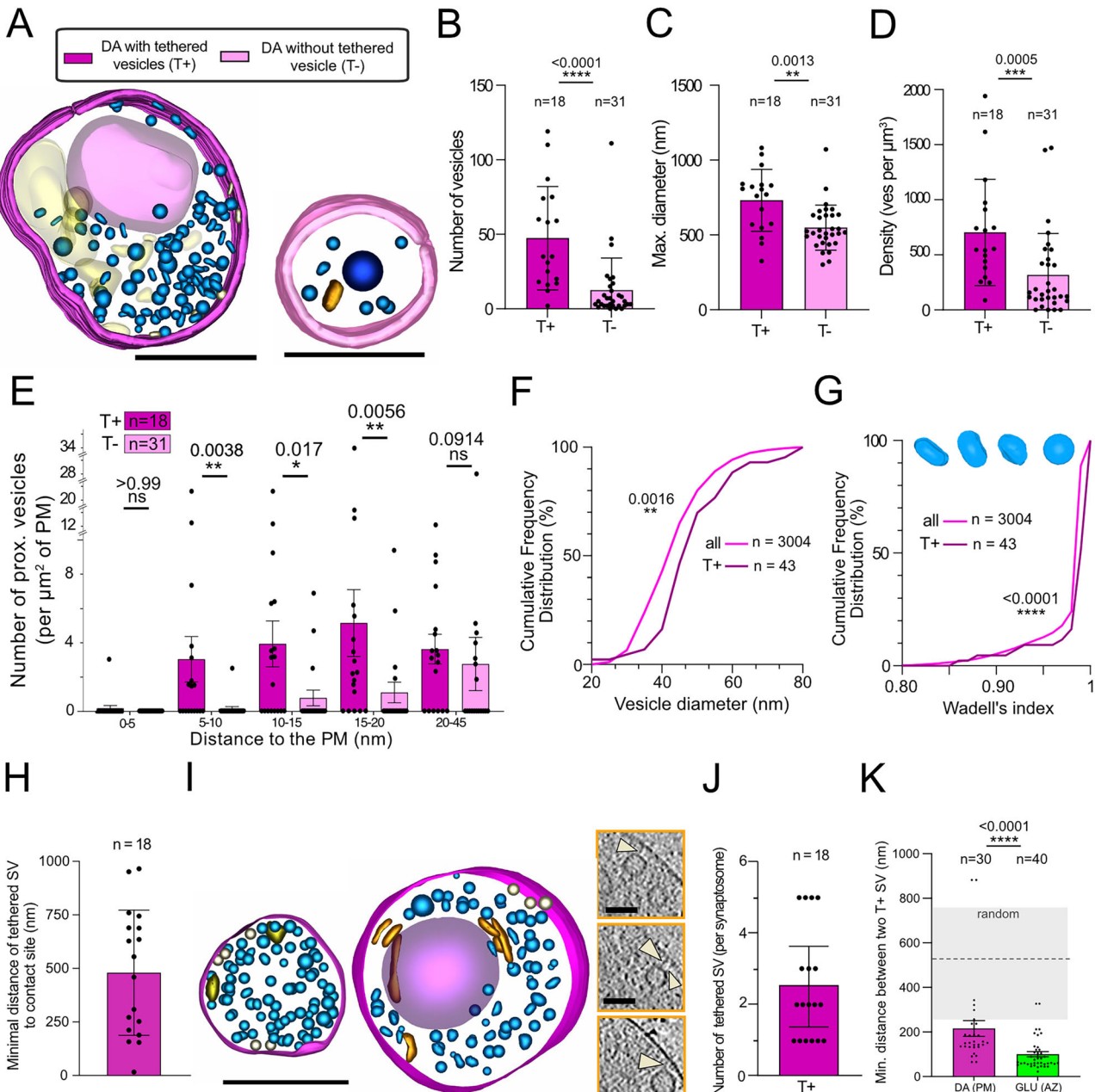

**Fig. 5 | Distribution of tethered vesicles in DA synaptosomes. A** 3D model of DA synaptosomes with (T, left) and without (T−, right) tethered vesicles. Scale bars, 500 nm. Color coding of organelles as in Fig. 1. **B** Total number of vesicles in DA synaptosomes with (T+) or without (T−) tethered SV (two-tailed Mann−Whitney; $p$ value < 0.0001). **C** Diameter of DA synaptosomes with (T+) or without (T−) tethered vesicles (two-tailed Mann−Whitney; $p$ value = 0.0013). **D** Density of vesicles in T+ vs T− synaptosomes (two-tailed Mann−Whitney; $p$ value = 0.0005). **E** Number of vesicles per μm² of plasma membrane at increasing distances from the plasma membrane for T+ and T− DA synaptosomes (Two-tailed $t$ tests). **F** Cumulative distribution of the vesicle diameter for tethered SV (T+) and all DA vesicles (Two-tailed Kolmogorov−Smirnov test). **G** Cumulative distribution of vesicle sphericity (WI) for tethered vesicles (T+) and all DA vesicles (Two-tailed Kolmogorov−Smirnov test). We show examples of DA vesicles with WI = 0.85, 0.89, 0.95, and 0.99 from

left to right above the plot. **H** Minimal distance along the plasma membrane between a tethered DA SV and the contact site with GLU presynapse ($n$ represents each tethered SV). **I** 3D models of two T + DA synaptosomes with tethered vesicles (in gray). The one on the left has five tethered vesicles on two sides of the terminal. The one on the right has a single cluster of three tethered vesicles on one side of the plasma membrane. Scale bars 500 nm. Right, corresponding tomogram images of the synaptosome with the three tethered vesicles. Scale bar 50 nm. **J** Number of tethered vesicles per T + DA synaptosome. **K** Average nearest neighbor distance between tethers on the plasma membrane for T + DA and GLU synaptosomes (two-tailed Mann−Whitney test). The distance between two random points is shown by the gray shaded area. Bar charts represent mean values ± SD (**B**–**D**, **H**, **J**) and ± SEM (**E**, **K**).

Synaptosomes obtained from the mouse brain constitute a reliable model to investigate the spatial configuration of synapses[17,39]. They are particularly amenable to cryo-ET because they can be directly observed by cryo-EM without further processing, such as cryo-sectioning or cryo-focused ion beam milling[17]. Moreover, they retain functionality, such as SV exocytosis and endocytosis, protein composition and post-synaptic calcium signaling[3,31,37,39]. Likewise, we show with FM4-64 labeling that our preparation of synaptosomes undergoes stimulation-dependent SV cycling. Subsets of both GLU and DA synaptosomes contain mitochondria (21/103 and 9/110, respectively).

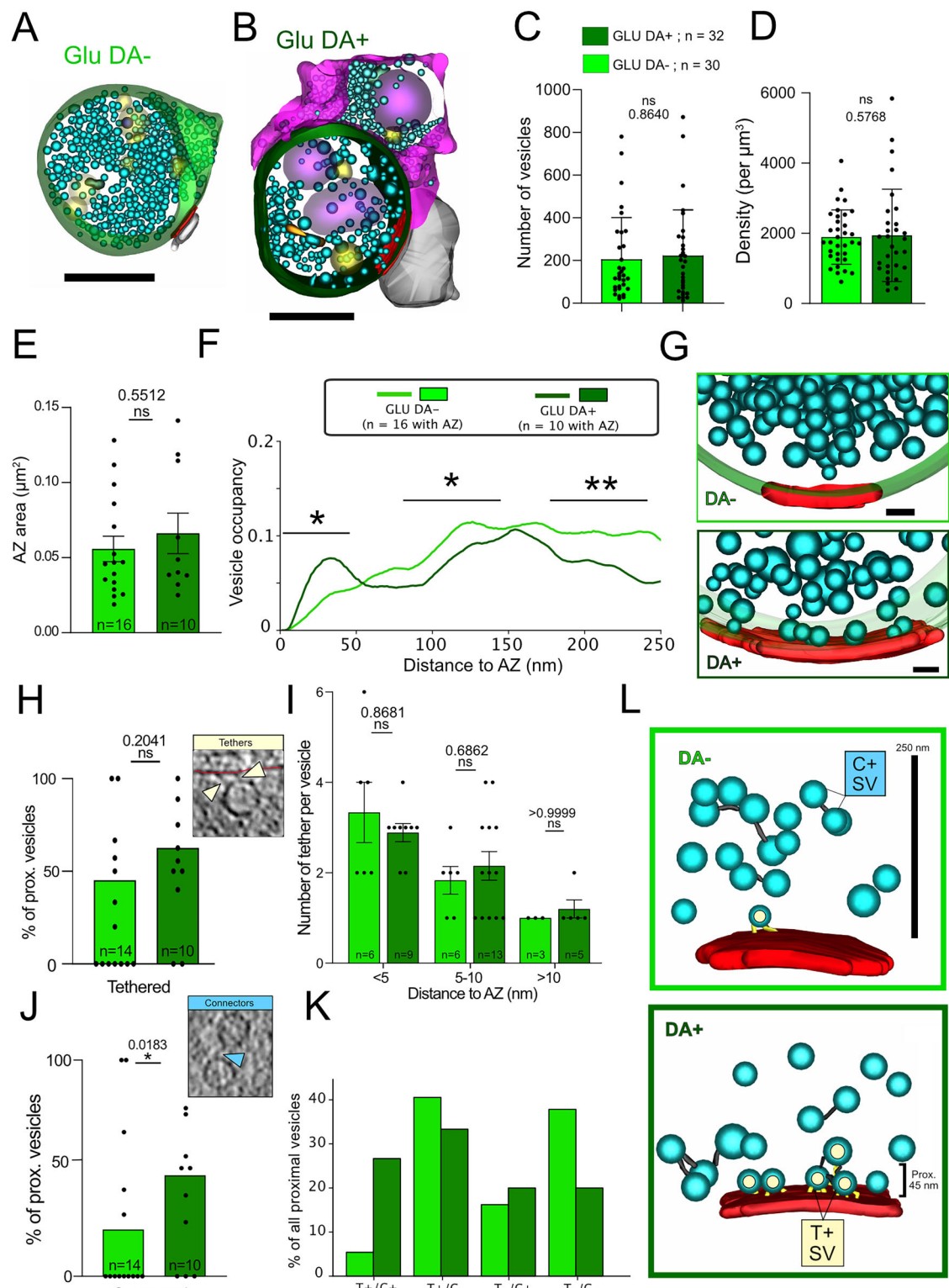

**Fig. 6 | Comparison of GLU synaptosomes connected or not to a DA terminal.**
**A**, **B** 3D models of a DA− GLU synaptosome (**A**) and a DA + GLU synaptosome or
DHS (**B**). Scale bars 500 nm. **C**, **D** Number of vesicles (**C**) (two-tailed Mann–Whitney;
*p* value = 0.864) and density of vesicles (**D**) (Mann–Whitney; *p* value = 0.577) in DA-
and DA + GLU synaptosomes. **E** Active zone area of DA- and DA + GLU synapto-
somes (two-tailed *t* test; *p* value = 0.551). **F** Average fraction of volume occupied by
SVs vs distance to the plasma membrane for DA− and DA + GLU synaptosomes
(two-tailed *t* tests). **G** 3D models of DA− and DA+ synaptosomes showing the
number of proximal vesicles at the active zone. **H** Percentage of proximal vesicles
(<45 nm from plasma membrane) that are tethered in DA− and DA + GLU

synaptosomes (one-sided chi² test; proportions for each synaptosome are shown as
data points, scale bar 50 nm). **I** Number of tethers per vesicle at various distances
from the active zone (two-tailed *t* tests). **J** Percentage of proximal vesicles that are
connected to other vesicles in DA− and DA + GLU synaptosomes (one-sided chi²
test; proportions for each synaptosome are shown as data points, scale bar 50 nm).
**K** Proportion of proximal vesicles that are tethered and/or connected in DA− and
DA + GLU synaptosomes. **L** 3D models of DA- and DA + GLU synaptosomes. In the
DA− GLU synaptosome, a single SV is tethered and not connected to other vesicles.
In the DA + GLU synaptosome, four SVs are tethered and connected to one more
vesicle. Bar charts represent mean values ± SD (**C**, **D**, **I**) and ± SEM (**E**).

These proportions are somewhat lower than the ones reported in synapses in tissue or cell culture (33.5% for GLU neurons in mouse cortex[49], 20–60% in cultured hippocampal neurons[50,51]; ~40% in DA neurons in striatum[16] and in culture[14]). Several factors might explain these differences. First, mitochondria may have been partially lost during synaptosome extraction, as they are often located at the periphery of axonal varicosities. Second, we excluded large synaptosomes, which are more likely to contain mitochondria, because they were too large to be imaged using cryo-ET. This exclusion could have introduced a bias into our sample. Despite these differences, the absence of mitochondria in synapses does not significantly impact the probability of SV release[51].

The GLU synaptosomes we obtained from the mouse striatum are qualitatively and quantitatively comparable to forebrain and hippocampal synaptosomes (likely GLU), and dissociated cultures observed previously with cryo-ET[3-5,31,32]. The presynaptic element contains hundreds of small round SVs (~40 nm diameter) and contacts a PSE with a clearly defined PSD and a 32 nm wide synaptic cleft filled with dense material. This defines an AZ of ~0.08 $\mu m^2$, in the range of reported sizes varying between 0.04 $\mu m^2$ and 0.10 $\mu m^2$[23,52,53]. Moreover, SVs are polarized towards the AZ, with a peak of volume occupancy ~25 nm from the plasma membrane, which reflects the enrichment in tethered and primed vesicles, similar to forebrain synapses and was proposed to be necessary for proper neurotransmitter release[54]. Nevertheless, this peak was less pronounced in our sample of CS synaptosomes than in forebrain synaptosomes[3,4], which may reflect genuine differences in relative SV pool sizes in cortical/hippocampal vs striatal synapses. Remarkably, proximal SVs located <5 nm from the AZ have in both CS and cortical synapses on average three tethers[4,5], which likely corresponds to the primed state of SV and to functionally defined readily releasable SVs. In neurons lacking Munc13-1 and Munc13-2, which abolishes priming[55], proximal SVs have only one tether[4]. Therefore, we propose that CS synapses have primed vesicles with the same structural hallmark as the ones defined in hippocampal synapses.

The most striking feature of DA synaptosomes is the relative sparsity of small synaptic-like vesicles, about 3-fold less dense than in GLU synaptosomes. Because DA synaptosomes are also smaller than GLU synaptosomes, there are 10 times fewer vesicles in DA synaptosomes than in GLU synaptosomes. These observations are consistent with ultrastructure of DA axons, determined with serial electron microscopy reconstruction in chemically fixed striatum tissue, in which axonal varicosities display very heterogenous ultrastructure: some have only vesicles, others have small synaptic-like vesicles, larger ones, or both types of vesicles[16]. Likewise, recent studies have shown qualitatively similar results in cultured DA neurons observed with cryo-CLEM[56] and DA neurons derived from human induced pluripotent stem cells[57]. Vesicles in DA synaptosomes are also significantly more elongated than vesicles in GLU synaptosomes. Elongated-shaped vesicles have been documented in DA axons[16,58] and at GABAergic inhibitory synapses with conventional electron microscopy[59] and also with cryo-CLEM[32]. Moreover, SVs in GLU synapses become elongated in the absence of VGLUT1, which reflects a lower luminal osmotic pressure[60,61]. Therefore, DA vesicles, like GABA vesicles, may experience a different osmotic pressure than GLU vesicles.

Importantly, we show that, similar to GLU synaptosomes, vesicles in DA synaptosomes are frequently linked together by connectors and to the plasma membrane by tethers. Nevertheless, only 37% of the DA synaptosomes contain at least one tethered vesicle. The DA synaptosomes with at least one tethered vesicle are bigger and contain twice as many vesicles as the ones without any tethered vesicles. Interestingly, other studies have shown that only ~30% of DA axonal varicosities contain AZ proteins bassoon, RIM or ELKS[12,14], and this proportion is also found for functional DA varicosities able to release fluorescent dopamine analog[15]. Therefore, we propose that synaptosomes containing tethered vesicles correspond to DA terminals with a higher

probability of dopamine release. Moreover, the number of tethers linked to proximal vesicles (less than 45 nm from the plasma membrane) is the same in GLU and DA synaptosomes. This suggests that similar molecules are involved in vesicle tethering in both types of synapses. Indeed, proteomic analysis of striatal DA synapses shows a similar degree of enrichment of the major proteins involved in vesicle tethering[27,62]. However, DA synaptosomes lacked very proximal vesicles (<5 nm from the plasma membrane), which have, on average, 3 tethers, a hallmark of vesicles primed for release in GLU synapses. This difference points to a fundamental difference between these two types of terminals. We thus predict that the kinetics of vesicle exocytosis will be much slower in DA neurons than in GLU neurons, where SV exocytosis occurs in less than 1 ms after calcium entry. So far, the kinetics of DA vesicle exocytosis have been determined with fast amperometry or fluorescence imaging of dopamine sensors[7,8]. The rise time of these events is greater than 10 ms, but it could be dominated by the diffusion of ligand to the detector micrometers away from the dopamine release site. More precise investigation of dopamine release is needed to determine if the kinetics of dopamine release are genuinely slower, and whether this is due to differences in vesicle priming.

We found a clear PSE at only 2 out of 110 DA terminals. These are among the synaptosomes containing many small round vesicles, and are thus indistinguishable from GLU synaptosomes. They may correspond to terminals of DA/GLU neurons co-expressing the vesicular glutamate transporter VGLUT2 in vivo[42,63,64] and also in cultured DA neurons[65]. In the adult mouse, these terminals are mostly concentrated in the shell of the nucleus accumbens[64]. Interestingly, in these axons, VGLUT2 is segregated from the vesicular dopamine transporter VMAT2, suggesting that glutamate and dopamine release sites are segregated[63]. In the other 108 DA terminals reconstructed, no clear PSE was identified.

We identified previously that DA terminals can strongly interact with other synaptic elements in a so-called DHS[27]. In the 32 reconstructed DHS, we found that 28 directly adhere to a presynaptic VGLUT1 element and 4 contact the GLU PSE. On the other hand, DA terminals, identified in conventional transmission EM with immunolabeling of tyrosine hydroxylase, are as likely to contact the pre- or post-synaptic side of CS synapses[24]. The selection of DHSs by cryo-fluorescence with the presynaptic marker VGLUT1-Venus likely biased our sampling towards interaction with presynaptic markers. Nevertheless, we could characterize both pre- and post-synaptic DA/GLU adhesion sites in their native state with cryo-ET. We found that the adhesion sites had similar areas, ~0.05 $\mu m^2$, as well as the cleft sizes, around 12 nm. However, these adhesions did not define a privileged tethering site for DA vesicles. Overall, we did not identify a localized site for vesicle tethering, suggesting a different organization for vesicle tethering in DA and GLU synapses. Nevertheless, we found that in DA synaptosomes with multiple tethered vesicles, these tended to cluster, albeit with larger distances between vesicles than in GLU synaptosomes. This suggests that, even in the absence of a clear morphological hallmark, molecular factors such as AZ proteins could direct vesicle tethering and fusion at specific sites in the DA varicosity.

Finally, we showed that the morphology of GLU terminals and the overall distribution of SVs are similar in GLU DA+ and GLU DA- terminals. Also, we detected tethers of different lengths, which were previously associated with different molecular compositions of tethers[3], in both types of presynaptic terminals. This suggests that the GLU terminal formation and its protein composition do not depend on the presence of DA terminals. Nevertheless, the distribution of proximal SVs in GLU DA+ terminals was consistent with those of non-perturbed GLU synapses, while GLU DA- terminals showed a flat proximal SV distribution profile previously associated with a reduced neurotransmitter release[3,4]. Furthermore, we correlated these differences with changes in SV connectivity and possibly also with tethering. Therefore, our data indicate that the contact between DA and GLU

terminals at DHS modulates the SV organization at the single-nanometer scale, and that this modulation is mediated by SV connectors. We could thus hypothesize that the formation of DHSs in the striatum is an important feature modulating glutamate release and striatal activity in vivo. These multipartite assemblies offer a close proximity between dopamine release sites and specific GLU synapses. By this mechanism, the spatial and temporal synchronization between dopamine and glutamate activity would be maximal, which could play a major role in the plasticity of excitatory input to striatal neurons[26].

## Materials and methods

### Animal models

We have used three mouse models. First, the VGLUT1[venus] knock-in mouse line in which CS synapses are labeled, and second, the DAT-cre BAC transgenic mouse line[35] in which we transduced VTA/SNc neurons with an AAV1 carrying pCAG-Flex-mNeongreen coding sequences to label DA neurons projecting to the striatum (coordinates from bregma are A/P: 2.9 mm; M/L: 1.6 mm; D/V: 4.6 mm with 12° angle). Finally, to co-detect CS synapses with DA terminals in the same sample, we used DAT-cre crossed with the reporter Ai14TdTomato mouse line, in which DA terminals are bright red, crossed with VGLUT1[venus] KI mice. We used adult, 12–18-week-old mice of both sexes.

We refined the experimental design and the procedures to reduce as much as possible the number of animals used and their suffering. All procedures were in accordance with the European guide for the care and use of laboratory animals and approved by the ethics committee of Bordeaux University (CE50) and the French Ministry of Research under the APAFIS no. #21132 and #38144.

### Preparation of synaptosomes

The preparation of synaptosomes was adapted from a previously published protocol[66]. Briefly, animals were euthanized by cervical dislocation, decapitated and the head was immersed in liquid nitrogen for 5 seconds for rapid cooling but not freezing of the tissue. The striata were subsequently dissected under an epifluorescence stereomicroscope (Leica Microsystems, Germany). Samples were then homogenized in 1.5 ml of ice-cold isosmolar buffer (0.32 M sucrose, 4 mM HEPES pH 7.4, protease inhibitor cocktail Set 3 EDTA-free (EMD Millipore Corp.)) using a 2-ml-glass-Teflon® homogenizer with 12 strokes at 900 rpm. The homogenate (H) was centrifuged at $1000 \times g$ for 5 min at 4 °C in a benchtop microcentrifuge. The supernatant (S1) was separated from the pellet (P1) and centrifuged at $12,500 \times g$ for 8 min at 4 °C. The crude synaptosomes pellet (P2) was suspended in 350 μL of isosmolar buffer and layered on a two-step ficoll density gradient (5 mL of 13% Ficoll and 5 mL of 7.5% ficoll, both in 0.32 M sucrose, 4 mM HEPES). The gradient was centrifuged at $50,000 \times g$ for 1 h and 10 min at 4 °C (Optima L100XP Beckman Coulter with SW32Ti rotor). The synaptosome fraction was recovered at the 7.5 and 13% ficoll interface using a 0.5 ml syringe. An additional centrifuge of the collected fraction was performed in 2 ml of HEPES-buffered Krebs-like solution (HBK)[38] composed of: 143 mM NaCl, 4,7 mM KCl, 1,3 mM $MgSO_4$, 1,2 mM $CaCl_2$, 20 mM HEPES, 0,1 mM $Na_2HPO_4$ and 10 mM D-glucose, pH = 7,4) at $12,500 \times g$ for 5 min in order to wash the excess of ficoll and sucrose residues. The barely visible pellet was resuspended in 200 μl of HBK leftover and placed for 15 minutes at 37 °C for physiological recovery before plunge freezing.

Alternatively, the S1 fraction diluted in HBK has been used, allowing for faster preparation with comparable results.

### FM4-64 uptake and release assay

To assess synaptosome integrity, we performed a fluorogenic vesicle exo-endocytosis assay using the amphiphilic styryl dye FM4-64 (Thermo Scientific T13320). Synaptosomes were prepared fresh with the same protocol as for cryo-CLEM. Then they have been diluted in HBK and centrifuged 34 min at $6750 \times g$ on 12 mm coverslips coated with 1 mg/ml Poly-L-Lysine. Imaging was performed right after on a wide-field epifluorescence microscope, Leica DMI8 equipped with an inverted ×63/1.4 oil objective, a Hamamatsu Flash 4.0 v2 camera and a controlled 37 °C/$CO_2$ chamber. Coverslips were placed in a recording chamber (Ludin), incubated with 500 μl of prewarmed HBK and imaged in TRITC and GFP channels at different registered positions over a 5 μm stack (1st acq). HBK was removed, and FM4-64 (6 ng/μl final concentration), KCl (40 mM final concentration) and 37 °C/$CO_2$ HBK mixture were added, inducing exo-endocytosis cycle and staining the membrane and internalized vesicles (see Supplementary Fig. 1 A). After 1 min 30 s, KCl was rinsed twice with HBK and FM4-64 with HBK was added, staining the remaining plasma membrane. After 1 min 30 s,the second acquisition was launched (2nd acq). After five rinses with HBK, a third acquisition captured signal from internalized vesicles only (3rd acq). KCl-HBK mixture was added to induce depolarization and exocytosis, and distaining was imaged 1 min 30 s after (4th acq). Loading was measured by subtracting the FM intensity of a single synaptosome at acquisition number. 3 (vesicles stained) versus acquisition no. 1 (background). Release was measured by subtracting the intensity of a single synaptosome at acquisition number. 3 (vesicles stained) with acquisition no. 4 (non-exocytosed vesicles and background dye binding). Control experiments with no KCl for loading or no KCl for release were performed. Analysis was performed on Fiji[67] using a home-built macro-command.

### Plunge-freezing

Quantifoil R2/2 Cu 200 or R3.5/1 Cu 200 mesh grids (Q-R2_2-2C100 or Q-3.5_1-3C100 Delta Microscopies) were glow-discharged for 30 seconds at 2.5 mA using an ELMO glow discharge system (Cordouan Technologies) to enhance their hydrophilicity. Following this, 4 μl of freshly purified synaptosomes, mixed with with 10 nm colloidal gold beads (Sigma 741957) and 100 nm FluoSpheres™ beads (F8797 ThermoFischer) were applied to the grids. The excess sample was immediately blotted away from the opposite side for 5 seconds in a Vitrobot Mark IV (Thermo Fisher Scientific), maintained at a temperature of 4 °C and at 100% humidity. Finally, grids were plunge-frozen in liquid ethane and stored in liquid nitrogen until observation.

### Cryo-fluorescence microscopy

We used a commercially available system to perform cryo-fluorescence microscopy (Leica – DM6 FS Cryo-CLEM). We monitored on test grids with TetraSpecks™ beads (Thermo Scientific T7279) shifts between channels and drift artifacts in cryogenic conditions. No significant shift was noticed, while noticeable mechanical drift occurs for acquisitions of more than 20 z-steps, which may be necessary when the grid is not flat and exactly parallel to the focal plane. Therefore, we limited our acquisitions to flat areas and imaged consecutive z planes with an increment of 0.5 to 1 μm (20 z-steps max). A custom-made Plexiglas® chamber was built around the setup to maintain relative humidity under 40% reducing contamination by water. The temperature of the objective chamber was maintained at −190 °C (83 K) using a pump projecting vapor of fresh liquid nitrogen. Each grid was observed separately, reducing the time spent in the chamber to 25 minutes maximum. Sample was observed with a ×50 objective (Leica – HC PL APO ×50/0.90NA Dry 11566064) and z-stacks were acquired through a Hamamatsu ORCA-Flash 4.0 camera, in four channels, in the following order: Brightfield (Empty); Tx Red (Em: BP560/40 nm; Ex: BP630/75 nm); GFP (Em: BP470/40 nm; Ex: BP525/50 nm); DAPI (Em: BP360/40 nm; Ex: BP470/40 nm). Large, flat areas of the grids were acquired using the LAS X Navigator mosaic imaging tool. Grids were stored in liquid nitrogen until the cryo-EM session.

Images from regions of interest were stacked in a maximal intensity Z-projection using Fiji software for each fluorescence channels[67]. Minimal Z-projection was performed for the brightfield channel, allowing the detection of holes in the carbon layer. Contrast

has been adjusted and allowed detection of all synaptosomes (dim and bright), as well as fluorescent beads. TIFF images have been saved in PNG file format for editing. Each square of interest (with one or more synaptosomes in a hole) was identified and marked with a unique number. Cropping of each corresponding area into a single image was done, and the resulting PNG files were saved-back into TIFF files for compatibility with Serial EM[68].

### Correlation between fluorescence and electron microscopy

The correlation process was carried out in two stages. The initial "approximate" correlation focused on locating the grid squares captured through cryo-fluorescence microscopy. After inserting the grid into the cryo-transmission electron microscope, a low-magnification montage (×34) was constructed using SerialEM[68] to visualize the entire grid. The fluorescence image was then imported into the software via the "Import Map" function. Areas of interest were identified using obvious landmarks, such as carbon holes and the center of the grid, which were visible in both imaging modalities.

Once the squares were identified, a high-magnification montage (×1600) was created for each area of interest. The corresponding fluorescence image for each square was imported separately. Precise correlation in SerialEM was done using the fluorescent fiducial beads as registration points, detectable in both fluorescence and electron microscopy. Beads were selected and assigned specific identifiers on the fluorescence image, and the same beads were subsequently located on the electron microscopy image. A total of 5–10 surrounding beads were used for the transformation to correlate with a single target for tomography. Both images were then correlated using the "Transform Item" function.

### Cryo-electron tomography

Tilt series of the synaptosomes were recorded using two different microscope setups. The first dataset was collected with a Talos Arctica microscope (Thermo Fisher Scientific), operating at 200 kV and equipped with a K2 Summit direct electron detector (Gatan). On this system, tilt series were acquired in a bidirectional scheme using SerialEM 4.0, covering angles from −60° to +60° with 2° increments. Images were recorded at a magnification of ×11,000 (pixel size of 3.987 Å). and a defocus of −8 μm, and a total electron dose of 80–90 e⁻/Å². These tilt series were subsequently processed using the IMOD software package 4.11.25[69] integrated into the SCIPION framework[70], with reconstructions achieved via weighted back projection and a SIRT-like filter. For initial screening data were also processed with EMAN2[71]. Final tomograms from this dataset were binned by 4, resulting in a pixel size of 1.595 nm.

A second dataset was acquired on a 200 kV Glacios 2 microscope (Thermo Fisher Scientific) equipped with a Falcon 4 camera (Thermo Fisher Scientific) and a Selectris X energy filter (Thermo Fisher Scientific) set to 10 eV. On this instrument, dose-fractionated movies were recorded in EER format using a dose-symmetric scheme in SerialEM (v. 4.1.21) from −60° to +60° with 2° increments, at ×39,000 magnification (3.117 Å pixel size) and a defocus of −4 μm, with a total electron dose of 90–100 e⁻/Å². This second dataset was processed using the RELION-5 pipeline[72], where raw movies were motion-corrected using RELION's implementation of MotionCor2[73] and CTF estimation was performed with CTFFIND4 (v. 4.1)[74]. Tilt series were aligned with AreTomo[75], and tomograms were reconstructed using RELION's own implementation. The final tomograms were binned by 4, resulting in a pixel size of 1.247 nm.

### Segmentation and production of models

We selected tomograms with accurate correlation, sufficient contrast and intact synaptic features for segmentation (numbers provided in Table S1). Other tomograms were discarded (examples in Fig. S8).

Manual segmentation was performed using 3dmod, a software from the IMOD package[76]. Synaptosome plasma membranes were segmented until disappearance and meshed; therefore, no interpolation was used and the missing wedge volume was not corrected for the synaptosomes. Organelles and vesicles were interpolated using spherical interpolation, resulting in closed objects. The AZ area was defined as the plasma membrane portion facing the PSD. Contact area (DHS) was the plasma membrane portion of the DA synaptosomes in tight apposition (<15 nm) to the GLU plasma membrane. SVs were defined as objects smaller than 80 nm with round or elongated shapes. Mitochondria were easily detected as dense, folded membranes present in the lumen. Vesicular bodies were observed thanks to the presence of one or more vesicle-like structures inside. Objects with folded, irregular shapes and larger diameters have been categorized separately and correspond to unidentified structures.

### Data analysis

We obtained geometrical information such as volume and maximum diameter from the 3D models using *imodinfo* command lines (IMOD package). Sphericity was calculated using Wadell's index[77] ($W_i$) with the following formula $W_i = \pi^{\frac{1}{3}}(6V)^{\frac{2}{3}}/S$ where $V$ represents the volume and $S$ the surface of the vesicle. Thus $W_i = 1$ for a perfect sphere and decreases for non-spherical objects.

Vesicle density inside synaptosomes was obtained by dividing the number of vesicles by the volume available inside the synaptosome. In Fiji, cytoplasmic density profiles were obtained by averaging a 40 nm stack and pixel intensities were measured using the built-in commands. In Fiji, synaptic cleft and interspace density profiles were measured at three different single z planes (low, middle and high) in the stack and averaged for each tomogram. Values were normalized to the neighboring non-cleft area corresponding to the background. We excluded tomograms in which gold beads were present, as they artificially alter the density.

SV tethers and connectors were detected in an automated, template-free manner using the hierarchical connectivity algorithm, and their morphology, localization and interrelationship were analyzed by Pyto package (version 1.10, available at https://github.com/vladanl/Pyto), as described before[31,48]. Briefly, for the analysis of vesicle distribution (volume occupancy), the presynaptic cytoplasm (including SVs) was divided into 1-pixel-thick layers according to the distance to the AZ membrane, and the fraction of the layer volume occupied by SVs was measured. Connector and tether lengths were calculated as the minimal edge-to-edge distance between connector/tether voxels contacting an SV or plasma membrane that takes into account central regions of tethers and connectors. In this way, the ambiguity inherent to the measurement of the length of 3D objects is resolved, and the curvature of tethers and connectors contributes to their calculated lengths. However, possible extended protein-lipid binding regions are not considered. All image processing and statistical analysis software procedures were written in Python and implemented in Pyto package. Pyto uses NumPy and SciPy packages and graphs are plotted using Matplotlib[78–80]. Statistical analysis was performed between the experimental groups using only planned, orthogonal comparisons. For the analysis of properties pertaining to individual SVs, connectors and tethers (such as the SV distance to the AZ membrane, tether length and fraction of tethers/connectors having a certain property), values within experimental groups were combined. Bars on the graphs show mean values and error bars the standard error of the mean (sem). In cases where a fraction of SVs or tethers is shown, the error bars represent sem between synapse means. We used Student's *t* test for statistical analysis of values that appeared to be normally distributed (e.g., vesicle diameter) and K−W test (nonparametric) for values deviating from the normal distribution (e.g., number of tethers and connectors per vesicle). For frequency data (e.g., fraction of connected and non-

connected vesicles), chi-square test was used. In all cases, confidence levels were calculated using two-tailed tests. The confidence values were indicated in the graphs by a single asterisk for $P < 0.05$, double for $P < 0.01$, and triple for $P < 0.001$. All values of statistical tests are presented in Table S3.

We focused on tomograms originating from the VGLUT1[venus] x DAT-cre x Ai14 tdTomato because it is the only model where we could reliably identify both GLU and DA and thus determine CS-DHS and non-CS-DHS conditions with certainty (see Table S2). Moreover, we selected tomograms where the contrast was optimal and sufficient information was available. We picked GLU synaptosomes where we identified the AZ, and for DA, where there were more than two vesicles. Thus, it ensures an optimal detection and a reliable comparison of the filaments between both conditions. We normalized tomogram density and applied a Kernel Gaussian filter (sigma = 2) to improve detection. Tomograms used for the GLU AZ were cropped on the dedicated region, and analysis was performed blind to the DHS, non-DHS condition.

### Estimation for shortest distance between tethered vesicles
We measured the arc distance between two tethered DA vesicles $d$ (that is the distance while remaining in the plasma membrane) by measuring the distance in the projection in the tomogram plane measured along the arc of the plasma membrane ($d_{xy}$) and the distance between the two tomogram planes in which the tethers connect the plasma membrane ($d_z$) as $d = \sqrt{(d_{xy})^2 + (d_z)^2}$. We estimated the distance between random points by taking the average maximum diameter of T + DA synaptosomes $D$ (722 ± 205 nm, Fig. 5C). For the distance between random points, we approximated the projection of synaptosomes by a circle of radius $R = D/2 = 361$ nm. The average arc distance between random points in a circle is half the length of a half circle (by symmetry), or $\frac{\pi}{2}R = 567$ nm and its variance is $\frac{\pi^2}{12}R^2$ that is a standard deviation of 327 nm. This estimate is a lower limit of the true value, because it neglects the axial distance $d_z$ and approximates synaptosomes as circles, minimizing distance.

### Reporting summary
Further information on research design is available in the Nature Portfolio Reporting Summary linked to this article.

## Data availability
Source data are provided with this paper. Binned tomograms and 3D models are available on EMPIAR. Source data are provided with this paper.

## Code availability
The code used in this study is available at https://github.com/vladanl/Pyto.

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

## Acknowledgements

We thank Peter Vanhoutte and Nicolas Heck for providing the DAT-cre * Ai14 tdTomato line, the Pôle in Vivo for animal breeding, husbandry, and help with stereotaxic injections. We thank members of the Bordeaux Imaging Center, Monica Fernandez-Monreal, for help with cryo-fluorescence microscopy and Fabrice Cordelières for the analysis of FM4-64 loading experiments. We thank student interns Timothé Lapha, Jade Giraud and Solène Hospital for the reconstruction of some of the tomograms. This work was supported by ANR (DopamineHub ANR-19-CE16-0003 to E.H. and D.P., FrontoFAT ANR-20-CE14-0020-03 to E.H. and UltraDopa ANR-24-CE16-5973-01 to E.H., D.P. and R.F.), the Fondation Recherche Médicale (to P.L., end of PhD and D.P., FRM team EQU202403018112), the European Research Council (ERC consolidator Grant PneumoTransfo to R.F.) and the Regional Council of Nouvelle Aquitaine (ParkSynGraft 205024) to E.H. and D.P, and by the French Government through the France 2030 program [grant number 21-ESRE-0024] managed by the French National Research Agency (ANR) as part of the "Investissements d'avenir" program.

## Author contributions

P.L. performed stereotaxic AAV injections. synaptosome preparations and live imaging of FM4-64 uptake with the help of V.P.-B. P.L. performed synaptosome freezing, cryo-fluorescence imaging, cryo-EM and cryo-ET with the help of R.A. and E.M. R.F. supervised the cryo-ET methodology. P.L. did all reconstructions and annotations of tomograms under the supervision of E.H. and D.P. P.L. performed quantitative analysis of all data together with V.L. P.L., R.F., E.H., and D.P. acquired funding. P.L., E.H., and D.P. wrote the manuscript, and all other authors edited it.

## Competing interests

The authors declare no competing interests.
