## [Transparent Peer Review file · Nature Communications]

Cryo-correlative light and electron tomography of dopaminergic axonal varicosities reveals non-synaptic modulation of cortico-striatal synapses

Corresponding Author: Dr David Perrais

Version 0:

Reviewer comments:

Reviewer #1

(Remarks to the Author)

In this manuscript, Lapios and colleagues used cryo-correlative electron tomography to examine the ultrastructure of glutamatergic and dopaminergic axonal varicosities in a preparation of synaptosomes isolated from the mouse striatum. The work shows very nice high-resolution images and models of such release sites and represents a useful addition to the scientific literature. This work is timely because work published in recent years has started to provide a new look at dopaminergic neuron axonal release sites, revealing that they can be quite different in their structure and in their functionality compared to more broadly studied glutamatergic and GABAergic synaptic release sites. The work appears to have been well done and analyzed and represents a body of work that is complementary and extends previous transmission EM work carried out on fixed tissues. The work is descriptive, but its comparative aspect nonetheless helps to solidify hypotheses concerning the differences between dopaminergic release sites, which are mostly non-synaptic in their structure, to more classical glutamatergic synapses, that are typically exclusively synaptic. The most interesting observation that comes out of the work is that axonal varicosities established by dopaminergic neurons appear to typically contain far fewer vesicles compared to those established by cortico-striatal glutamatergic release sites. I find the discussion of the manuscript to be the weakest part, as it fails to discuss some of the limitations of the work that was carried out. It also does not sufficiently clarify the key conceptual advances that derive from the work (and that were not already made previously with other approaches). As an example, it would have been good to see the authors reconsider the previous large body of work carried out on dopaminergic neuron release sites using transmission EM and fixed tissues.

Important issues to consider:

1. In the introduction, the authors allude to the possibility that perhaps only a small subset of axonal varicosities are functional. I would encourage the authors to be careful with such a statement. The techniques used to draw such a conclusion (refs 14 and 23 of the present manuscript) are limited by the sensitivity of the detection techniques used. If what the authors show in the manuscript is correct (that some varicosities contain small number of vesicles), it could very well be that the dopamine released by the smaller varicosities was under the threshold of detection and thus not detected by the techniques used. See also this manuscript for evidence that most dopaminergic axonal varicosities might in fact be able to undergo activity-dependent exocytosis: PMID: 34320240.
2. Similarly, in the introduction, the authors cite some work suggesting that dopamine only diffuses over 2µm once released from varicosities. This conclusion is here again completely dependent on the affinity of the sensor used. In the case of the carbon nanotube sensor that is mentioned, the affinity (Kd) appears to be 11 µM, which is quite low and would only detect large peaks of dopamine. I encourage the authors to consider this in their presentation and discussion of such data.
3. In the introduction, the authors write "Isolation of DA synaptosomes from striatal tissue and sorting by fluorescence activated synaptosome sorting (FASS) revealed that most DA terminals are in close contact with other terminals, such as GLU presynapses." I suggesting revising this statement to: "Isolation of DA synaptosomes from striatal tissue and sorting by fluorescence activated synaptosome sorting (FASS) revealed that most DA terminals isolated with this method, are in close contact with other terminals, such as GLU presynapses." This conclusion could be greatly influenced by the technique used.
4. In the results section, the authors refer to some of their results obtained with the activity-dependent dye FM4-64. They go on to cite two papers in support of their statement. It would be good to recognize that there were much earlier reports of the use of FM dyes with synaptosomes. For example PMID: 7912090.
5. The authors show that only 21 out of 101 glutamatergic varicosities contained a mitochondrion. For dopaminergic

varicosities, they report in table 2 that the proportion is even smaller (9%). These results are in contradiction with much of the previous literature, showing that in fact, a majority of axon terminals appears to contain mitochondria. For example, a recent study cited by the authors showed that in cultured DA neurons, most axonal varicosities contain mitochondria (<https://doi.org/10.1101/2024.04.15.589543>. See also: PMID: 36192156). One interpretation of this discrepancy is that the synaptosome isolation preparation method, perturbed the content of the collected axonal varicosities. It is important that the authors discuss this as it highlights a potential important limitation of the approach used.

6. Because the authors mentioned the proportion of Glu varicosities that contain mitochondria in the results section, this proportion also needs to be stated in the text for DA varicosities. Not just in Table 2.

7. The authors report in their description of figure 2 that dopaminergic axon varicosities contain pleiomorphic vesicles. This is something that was reported before. Please make sure to state this and cite prior work by Nirenberg and Pickel.

8. In their results section, the authors allude to the interesting possibility that the larger varicosities that also appear to form synapses are perhaps sites for glutamate release by dopamine neurons that express VGLUT2. This is certainly a possibility. The two papers that are cited to support this statement are not the most appropriate. The first reports of glutamate release by DA neurons and of VGLUT2 expression by these neurons are from the following two publications, which should be cited instead of the two that are presently cited. PMID: 15009640 PMID: 9614234.

9. On page 11 of the results section, the authors state that "The percentage of tethered vesicles among proximal vesicles in DA is $25 \pm 6\%$, which was lower, but not significantly different from GLU synaptosomes (Figure 4C)". The sentence should be modified to remove "which was lower". If the difference is not significant, it is not appropriate to mention that it was lower. Same problem for a sentence in the section describing the results of figure 6: "The fraction of tethered vesicles was about 70% bigger in GLU DA+ terminals but did not reach significance.". The sentence needs to be modified accordingly.

10. It is strange that in panel 4L, a t-test was performed to compare the proportions, but no error bars are shown. This needs to be clarified.

11. The section of the results linked to figure 5 is the weakest of the study. The main conclusions are based on only 6 dopaminergic synaptosomes that contained multiple tethered vesicles. The conclusion that the location where most of the tethered vesicles is a putative active zone is perhaps correct, but there is simply not enough data to support it. Globally, this figure does not add much to the manuscript and could easily be removed.

12. I don't think that it is appropriate to use multiple t-tests to compare the results shown in panel 6K because the data sets are linked. The authors should reconsider this and consult a biostatistician.

13. In the discussion, the authors write "Therefore, we propose that synaptosomes containing tethered vesicles correspond to release-competent DA terminals." This conclusion is not the only interpretation of such results. I invite the authors to consider other published results with higher sensitivity techniques suggesting that in fact, a majority of axonal release sites established by DA neurons are release competent, even if they do not express high levels of active zone proteins. See for example PMID: 34320240.

More minor issues:

1. On page 8, what the authors mean by «The normalized densities decrease by 20% in both compartments» is unclear.

2. In all bar graphs, the individual data points should be illustrated.

Reviewer #2

(Remarks to the Author)

What are the noteworthy results?

This work provides a comprehensive structural characterization of synaptosomes isolated from mouse striatum. The novel and noteworthy results arise from the isolation of dopamine hub synaptosomes which contain dopaminergic terminals in apposition to glutamatergic synapses, and the characterization of all structures involved.

- Will the work be of significance to the field and related fields? How does it compare to the established literature? If the work is not original, please provide relevant references.

This work provides structural characterization that could support future studies investigating the regulation of striatal glutamatergic synapses by dopamine release. This manuscript provides more comprehensive characterization of "dopamine hub synapses" that have previously been isolated and characterized by the same group.

- Does the work support the conclusions and claims, or is additional evidence needed?

Given that the manuscript is primarily descriptive in nature, there is not a strong claim. The data provided does support the hypothesis that dopamine hub synapses (or synaptosomes) are unique structures, there are differences between glutamatergic and dopaminergic terminals involved in the DHS, and that there may be small ultrastructural differences in GLU synapses that are contacted by DA terminals compared to those that aren't.

- Are there any flaws in the data analysis, interpretation and conclusions? - Do these prohibit publication or require revision?

There are several minor revisions that must be made to the manuscript before publication. See the attached list.

- Is the methodology sound? Does the work meet the expected standards in your field?

Yes. The study has been well-designed and clearly communicated.

- Is there enough detail provided in the methods for the work to be reproduced?

Yes. The methods are comprehensive and well-explained.

Suggested edits

New analyses:

- From the main text: “First, the active zone could locate at or right next to the contact zone with GLU terminals. However, tethered vesicles are almost never found at the contact site.” – quantify where the tethered vesicles are located relative to the contact site
 - o For example, analyze distance along the plasma membrane from the center of contact site to the vesicle tether
- Comment on the sphericity of tethered DA vesicles compared to the average sphericity for all vesicles (or all untethered vesicles)
- Two different labelling methods were used to identify and characterize GLU and DA synapses (AAV vs genetically encoded). It seems that synaptosomes from both labelling methods were pooled for analysis of the characteristics of individual DA and GLU synapses. If not, please specify in figure captions and in main text which method was used. If so, please provide evidence that there is no significant difference in characteristics (size, vesicle size and occupancy) between synapses from AAV labelled and Ai14-tdtomato labelled samples
- How often DA terminals appose a membrane that is not recognizable as a GLU terminal
- Quantify location of DA/GLU contact site compared to active zone – for example, analyze distance along the plasma membrane from the center of the active zone to the center of the contact site
- From text: “However, the absence of a morphologically identifiable DA active zone, may have biased the measurement of occupancy, because it included a larger, possibly irrelevant volume in our analysis.” This is an important point. One way to work around this bias is to perform similar analysis for GLU terminals. In other words, quantify the number of vesicles in GLU terminals that are proximal to the whole plasma membrane rather than just the active zone. Are there any instances of GLU SVs that are tethered outside the active zone? If so, how many?
- The minimum distance between tethered vesicles (along the plasma membrane) is quantified for DA synaptosomes. Please provide the minimum distance measurements for GLU synaptosomes also.

•
•

Main text revisions:

- Please include a citation for the following statement “Importantly, CS-DHS contain an increased signal for the presynaptic proteins VGLUT1 and bassoon compared to other CS synapses”
- From text: “The normalized densities decrease by 20 % in both compartments”
 - o When characterizing the electron densities/protein composition, it’s implied in the figure that this decrease is comparing cytoplasmic protein density to plasma membrane density but this is unclear in the main text. Please clarify this statement in the main text to what this decrease refers.
- “Another DA synaptosome has 295 vesicles (see two upper dots in Figure 2E) and also resembles classical GLU synaptosomes, even though it was not connected to a PSE but engaged into a CS-DHS”
 - o This sentence implies that the two dots refer to the DA synaptosome with 295 vesicles and no PSE but based on the figure, these two dots correspond to the DA terminals that have a PSE (with ~600 and 40 vesicles). Please clarify in the main text, add all 3 outlier points to the bar charts, and explain what the dots correspond to in the figure caption.

General revisions to manuscript content

- “Overall, we did not identify a localized site for DA vesicle tethering, suggesting a large AZ for exocytosis” – I don’t think this is an appropriate claim because this could also mean no “active zone” especially given the lack of evidence of any dopamine release in these synaptosome preps (in other words, there’s no evidence that the DA synaptosomes specifically are active).
- “Therefore, our data indicate that the contact between DA and GLU terminals at DHS modulates the SV organization at the single nanometer scale, as well as release properties of GLU terminals, and that this modulation is mediated by SV connectors.” – This is not an appropriate claim. There is no evidence of differential release properties of GLU terminals here – just slight structural changes.
- Not sufficient evidence for this statement: “Moreover, in T+ DA synaptosomes, vesicles are located closer to the plasma membrane than in T- DA synaptosomes (Figure 5G), reinforcing the idea that in DA synaptosomes vesicle location is polarized to a putative active zone where vesicle may tether and fuse”
- Therefore, we propose that synaptosomes containing tethered vesicles correspond to release-competent DA terminals – This is purely speculative as no data is provided that dopamine is actually released from these synaptosomes. Wished there was at the least Bassoon/RIM labelling to the DA synaptosomes, given that ~30% synapses that contain active zone proteins are the high Pr sites [PMC5807134].

•

Figure revisions

- Because this is primarily a descriptive publication, all figures and graphs would benefit from inclusion of individual data points rather than means along (relevant in figures...)
- Figure 1:
 - o A, right, bottom: What does the black dotted line correspond to?
 - o Please indicate in both D (right) and E (left) where the insets (panel E, right) align
- Figure 2:
 - o C-F: as mentioned above, please explain in figure captions what the green dots correspond to
- Figure 3:
 - o A-C: quantify, either in a figure inset or in the figure caption, what proportion of segmented synaptosomes correspond to each DA/GLU/PSE configuration (2/94 for C, for example)
 - o E: Does the % of cleft distance start from the presynaptic or postsynaptic neuron? Explain x-axis in figure caption
 - o D/E, H/I: Add to supplement - for DA/PSE contact, is there similar electron density between the two membranes?
- Figure 4:
 - o In main text: “The percentage of inter-connected SVs was significantly higher in GLU (53%) than in DA synaptosomes

(20%) (Figure 4J) and the length of connectors was significantly different (DA: 18.50 nm; std. 9.18 and GLU: 16.33 nm; std. 13.07; p-value = 0.0021) (Figure 4K).”, text does not match figure

o L: add error bars and individual points, as suggested above (what percentage per segmented synaptosome)

• Figure 5:

o B,D, F: normalize to synaptosome volume (in addition or instead of raw numbers)

o F: include a benchmark for the average distance to a random point on the membrane for comparison (as described in text)

• Figure 6

o K: add error bars, would also be interesting to see the n for each bin (n would be helpful in all bar graphs)

• Figure S1:

o these are not violin plots – but would benefit from being violin plots (see above about including the individual data point distributions)

• Figure S3/Table 2:

o Expand table 2 to contain the “other” segmented organelles, for GLU terminals, DA terminals, and any organelles present in PSEs

• Figure S9:

o G: add error bars

• Fig S10:

o A, C: Add individual values (per synaptosome segmented), error bars

o A-D: Mention whether any of these comparisons are significant, either on figure or in figure caption

o B: Function cumulates to 1 at only 20nm away from the plasma membrane, while proximal vesicles were defined as those within 45nm of the plasma membrane. Extend the x-axis to include all vesicles up to 45nm away from the membrane

• Table 1: This is a somewhat confusing format. It would be more understandable if the singly labeled samples were listed in a separate table from the dually labeled samples, possibly include a venn diagram for each DA/GLU synaptosome and whether it was involved in a DHS, the “N” in the dually labeled synaptosomes presumably means that, for example, DA synaptosomes not involved in DHS appeared in 8/10 preps, but this is confusing as the total number of preps doesn’t add up to 10. I don’t find this particularly informative, especially given that there’s no labeling of what each prep was (for example, there’s no indication of what the overlap is for each prep)

Reviewer #3

(Remarks to the Author)

Lapios et al. characterize the architecture of two major types of synapses found in dopaminergic neurons: glutamatergic and dopaminergic synapses. A key highlight of the study is the integration of cryo-electron tomography (cryo-ET) with fluorescence imaging to accurately depict striatal synaptosomes composed of dopaminergic (DA) nerve terminals.

This work aligns well with previous findings. First, from serial section EM of DA neurons in situ, which initially revealed vesicle size pleomorphism (PMID: 34965204), and later, from a study suggesting that such pleomorphism is inherent to the type of vesicular transporters involved (PMID: 39788994). This was observed across multiple systems, including in vitro reconstitutions, iPSC-derived DA neurons, and striatal synaptosomes.

Therefore, the publication of this manuscript is both timely and relevant to the emerging concept of non-canonical synapses, which likely represent the dominant mode of action in dopaminergic neurons. I am, thus, in favor of its publication, pending the authors’ response to some reviewer comments.

1) In Figure 1G, what is the morphological difference between SV (blue) and endosome-like vesicle (yellow)? Is it possible to show the sizes of them? Do endosome-like vesicle increase when DA and Glu synapses contact?

2) In Figure 2G, the authors show the cumulative frequency distribution of vesicle diameter. Can the authors also include a dot plot to better depict the individual sizes of VGLUT+ vs DAT+ vesicles?

3) The authors mention that 25% of the dopamine hub synapses (DHS) contact the cortico-striatal region. What about the remaining 75%, what do they connect to? Additionally, do dopaminergic neurons modulate only excitatory synapses, or do they also influence inhibitory ones?

4) I wonder if the size of the synaptosomes is also dependent on the presence or absence of mitochondria. Can a plot be shown for this feature?

5) How are inter-connected SVs measured, by physical distance or presence of connectors?

6) In Figure 5, do you think the distribution of more “flat” vesicles is dependent on their proximity to PM or completely independent of tethering properties?

7) Previous studies have shown that GABA (PMID: 23034651) can also be secreted from dopaminergic axons. Did the authors investigate VGAT-positive vesicles? Some GABA vesicles are known to have a more “flat” morphology.

8) If possible, it would be informative to assess the vesicle density on the plasma membrane during stimulation, particularly in dopamine varicosities. This could help determine whether the vesicles are functionally active or simply stored.

Reviewer #4

(Remarks to the Author)

Lapios and colleagues present work describe a series of studies that employ a combination of cryo-correlative light and electron microscopy (cryo-CLEM) and cryo-electron tomography (cryo-ET) in the context of imaging mouse striatal dopaminergic and glutamatergic synaptosomal preparations. Using these approaches, the authors describe the three-dimensional architecture of glutamate and dopamine synapses. This includes providing novel details that distinguish these

different families of synapses and their organization. Just as importantly, the manuscript employs ultrastructural reconstructions and thorough quantitation to define subpopulations of dopamine terminals according to vesicle tethering. Finally, this work analyzes differences between glutamatergic synapses that are connected to dopamine terminals versus those synapses that are not, offering structural details that further enhance the details structural characterizations of these synaptic structures.

Altogether, the above results presented in the manuscript are consistent with and build upon a robust earlier literature that has characterized both glutamatergic and dopaminergic synapses in mammals via room temperature transmission electron microscopy, serial volume electron microscopy, as well as as cryo-CLEM/cryo-ET to a lesser degree (e.g., PMID: 34965204, 29311144). Nevertheless, there are several key weaknesses that significantly diminish enthusiasm for the present work:

1) Foremost, the entire premise of the work is built around the use of purified synaptosomal preparations followed by fluorescence-based FACS sorting. Both of these methods are not gentle, but rather introduce substantial shearing forces to the structures being studied here - a fact that the authors acknowledge. Given the physical stresses introduced to the experimental system, and the fact that virtually all of the ultrastructural findings absolutely depend upon intact synaptic and vesicular structures, it is very difficult to rule out that the sample preparation may have distorted at least some of the reported findings. The ability to confirm the study's conclusions in a more native context would be strongly recommended. For example, the authors could conduct CLEM or serial EM (at room temperature or under cryogenic conditions) on intact cellular preparations examining these same dopaminergic and glutamatergic synapses.

2) Given the preponderance of heterogeneous vesicular structures in dopaminergic synapses, it is difficult to determine whether some of these structures are truly synaptic vesicles or other irregularly shaped organelles such as endosomes. This would be remedied by labeling of vesicles with a fluorescently-tagged vesicular monoamine transporter 2 (VMAT2) - a definitive marker of monoaminergic synaptic vesicles. Indeed, in one instance, the authors describe a dopamine synaptosome where most of the vesicles are small and round, which differs markedly from their other preparations, raising the possibility that this may not be a true dopaminergic synaptic structure. Having an additional tag like VMAT2 would increase the confidence in such a description.

3) Finally, one of the defining features of dopaminergic synapses is the reliance on bulk neurotransmission which differs substantially from glutamatergic neurotransmission, particularly at type 1 synapses. Having a more detailed discussion of these differences would clarify and inform the Discussion section.

Reviewer #5

(Remarks to the Author)

Version 1:

Reviewer comments:

Reviewer #1

(Remarks to the Author)

The revised manuscript corrects some of the issues raised in the first version. A few smaller issues still remain to be corrected:

1. Line 68: replace the word "terminal" by "varicosity" because a varicosity not containing any vesicles cannot really be called a terminal.
2. Lines 72-73: What the authors mean by "difficulties to identify rare DA axons" is unclear. DA axons in the striatum are not rare.
3. Lines 208-209: The sentence beginning with "Notably" is incomplete and needs to be revised.
4. As mentioned by the authors on lines 221-222, an active zone is typically defined, at least for glutamate and GABA synapses, as a membrane region in close contact with a postsynaptic membrane. Because the authors clearly show that this is very rare for DA neuron terminals, in keeping with much previous transmission EM work, do the authors conclude that DA neurons do not possess an active zone? In the discussion, the authors do refer to the idea that DA terminals appear to possess release sites that are different compared to glutamate terminals, but they do not make a very clear point about what they suggest. It would be good to clarify this.
5. Lines 226-227: the sentence is unclear. What the authors are trying to say by "membrane it in the pre-synapse" is confusing.

Reviewer #2

(Remarks to the Author)

The authors adequately addressed my concerns.

Reviewer #3

(Remarks to the Author)

The authors have adequately addressed my questions. I recommend publication of the manuscript.

Reviewer #4

(Remarks to the Author)

The authors have ably addressed all the points raised in the review.

Reviewer #5

(Remarks to the Author)

Answer to the reviewers' comments

Reviewer #1 (Remarks to the Author):

In this manuscript, Lapios and colleagues used cryo-correlative electron tomography to examine the ultrastructure of glutamatergic and dopaminergic axonal varicosities in a preparation of synaptosomes isolated from the mouse striatum. The work shows very nice high-resolution images and models of such release sites and represents a useful addition to the scientific literature. This work is timely because work published in recent years has started to provide a new look at dopaminergic neuron axonal release sites, revealing that they can be quite different in their structure and in their functionality compared to more broadly studied glutamatergic and GABAergic synaptic release sites. The work appears to have been well done and analyzed and represents a body of work that is complementary and extends previous transmission EM work carried out on fixed tissues. The work is descriptive, but its comparative aspect nonetheless helps to solidify hypotheses concerning the differences between dopaminergic release sites, which are mostly non-synaptic in their structure, to more classical glutamatergic synapses, that are typically exclusively synaptic. The most interesting observation that comes out of the work is that axonal varicosities established by dopaminergic neurons appear to typically contain far fewer vesicles compared to those established by cortico-striatal glutamatergic release sites. I find the discussion of the manuscript to be the weakest part, as it fails to discuss some of the limitations of the work that was carried out. It also does not sufficiently clarify the key conceptual advances that derive from the work (and that were not already made previously with other approaches). As an example, it would have been good to see the authors reconsider the previous large body of work carried out on dopaminergic neuron release sites using transmission EM and fixed tissues.

We thank the reviewer for his/her positive evaluation of our work. We have in particular extensively rewritten the discussion and we have addressed all his/her specific points.

Important issues to consider:

1. In the introduction, the authors allude to the possibility that perhaps only a small subset of axonal varicosities are functional. I would encourage the authors to be careful with such a statement. The techniques used to draw such a conclusion (refs 14 and 23 of the present manuscript) are limited by the sensitivity of the detection techniques used. If what the authors show in the manuscript is correct (that some varicosities contain small number of vesicles), it could very well be that the dopamine released by the smaller varicosities was under the threshold of detection and thus not detected by the techniques used. See also this manuscript for evidence that most dopaminergic axonal varicosities might in fact be able to undergo activity-dependent exocytosis: PMID: 34320240.

We have toned down this part of the Introduction. We now write: "Moreover, DA axons contain AZ proteins, RIM1, Munc13 and ELKS, which are important for fast dopamine release^{12,13}. However, only ~30 % of DA varicosities contain such assemblies^{12,14}. Intriguingly, uptake of fluorescent dopamine analogue in ex vivo striatal slices shows that only ~25% of all DA varicosities release it after electrical stimulation¹⁵, which suggests functional heterogeneity of these terminals." We now employ the term functional heterogeneity which does not imply necessarily that 75% of terminals would be inactive. Indeed, we agree that the quoted studies could have missed varicosities under the threshold of detection. Moreover, we now cite Ducrot et al. 2021 (PMID: 34320240, now ref 14) which also shows that DA varicosities vary in their content in the AZ proteins bassoon and Rim1/2 in cultured DA neurons.

2. Similarly, in the introduction, the authors cite some work suggesting that dopamine only diffuses over 2 μm once released from varicosities. This conclusion is here again completely

dependent on the affinity of the sensor used. In the case of the carbon nanotube sensor that is mentioned, the affinity (K_d) appears to be 11 μM , which is quite low and would only detect large peaks of dopamine. I encourage the authors to consider this in their presentation and discussion of such data.

We did not mean that dopamine will diffuse only over 2 μm . Our argument is that near putative release sites dopamine can reach higher concentrations. We have now rephrased this sentence as follows: "This proximity is functionally relevant, because DA release generates micrometer-wide hotspots of dopamine as estimated using carbon nanotube or genetically encoded dopamine sensors^{7,8,24}." In addition to the work with carbon nanotubes (ref 24), we now cite again work using genetically encoded sensors (ref 7 and 8) for which the authors detect hotspots with sensors of various affinities (EC_{50} of 10 nM for DA1h, 130 nM for DA1m and 330 nM for dLight1.1).

3. In the introduction, the authors write "Isolation of DA synaptosomes from striatal tissue and sorting by fluorescence activated synaptosome sorting (FASS) revealed that most DA terminals are in close contact with other terminals, such as GLU presynapses." I suggesting revising this statement to: "Isolation of DA synaptosomes from striatal tissue and sorting by fluorescence activated synaptosome sorting (FASS) revealed that most DA terminals isolated with this method, are in close contact with other terminals, such as GLU presynapses." This conclusion could be greatly influenced by the technique used.

We propose with the quoted sentence and the following one that if physical interactions between cells are conserved after applying shearing forces during tissue homogenization and FASS, then they must be specific and stable interactions at the time of animal death. Adding the words "isolated with this method" could suggest that homogenization and/or FASS could induce these interactions, which is highly unlikely because we provide several evidence in Paget-Blanc et al. 2022 (PMID: 35660742) that no aggregation of synaptosomes is observed after FASS using our method. We now also detailed our protocol in a new book chapter with specific notes regarding unspecific aggregates (Paget-Blanc et al., 2025). Therefore, we would like to keep the sentence as it was written.

4. In the results section, the authors refer to some of their results obtained with the activity-dependent dye FM4-64. They go on to cite two papers in support of their statement. It would be good to recognize that there were much earlier reports of the use of FM dyes with synaptosomes. For example PMID: 7912090.

We apologize to the reviewer for not citing the original article using FM1-43 to detect SV exocytosis in synaptosomes. We now cite it (now ref 38).

5. The authors show that only 21 out of 101 glutamatergic varicosities contained a mitochondrion. For dopaminergic varicosities, they report in table 2 that the proportion is even smaller (9%). These results are in contradiction with much of the previous literature, showing that in fact, a majority of axon terminals appears to contain mitochondria. For example, a recent study cited by the authors showed that in cultured DA neurons, most axonal varicosities contain mitochondria (<https://doi.org/10.1101/2024.04.15.589543>. See also: PMID: 36192156. One interpretation of this discrepancy is that the synaptosome isolation preparation method, perturbed the content of the collected axonal varicosities. It is important that the authors discuss this as it highlights a potential important limitation of the approach used.

With our new dataset (103 GLU synaptosomes and 110 DA synaptosomes) we keep the proportion of 20.4 % (21/103) and 8.2% (9/110) of GLU and DA synaptosomes containing mitochondria, respectively. Moreover, we now report in the Results that GLU and DA synaptosomes containing mitochondria are larger than the ones without (max extension 1045

± 205 nm vs 785 ± 250 nm, $p < 0.0001$ for GLU synaptosomes, and 838 ± 224 nm vs 555 ± 176 nm, $p < 0.0001$ for DA synaptosomes).

We now write in the Discussion:

“Subsets of both GLU and DA synaptosomes contain mitochondria (21/103 and 9/110, respectively). These proportions are somewhat lower than the ones reported in synapses in tissue or cell culture (33.5% for GLU neurons in mouse cortex⁴⁹, 20–60% in cultured hippocampal neurons^{50,51}; ~40% in DA neurons in striatum¹⁶ and in culture¹⁴). Several factors might explain these differences. First, mitochondria may have been partially lost during synaptosome extraction, as they are often located at the periphery of axonal varicosities. Second, we excluded large synaptosomes, which are more likely to contain mitochondria, because they were too large to be imaged using cryoET. This exclusion could have introduced a bias into our sample. Despite these differences, the absence of mitochondria in synapses does not significantly impact the probability of synaptic vesicle (SV) release⁵¹..”

We did not cite Lycas et al. bioRxiv (DOI 10.1101/2024.04.15.589543) because we have enough published data which was peer-reviewed that addresses the issue.

6. Because the authors mentioned the proportion of Glu varicosities that contain mitochondria in the results section, this proportion also needs to be stated in the text for DA varicosities. Not just in Table 2.

We now write in the Results: “DA synaptosomes containing mitochondria (9/110) are significantly larger than the ones who do not (max extension 838 ± 224 nm vs 555 ± 176 nm, $p < 0.0001$).

7. The authors report in their description of figure 2 that dopaminergic axon varicosities contain pleiomorphic vesicles. This is something that was reported before. Please make sure to state this and cite prior work by Nirenberg and Pickel.

We have changed the text in the Discussion to: “Vesicles in DA synaptosomes are also significantly more elongated than vesicles in GLU synaptosomes. Elongated-shaped vesicles have been documented in DA axons^{16,57} and at GABAergic inhibitory synapses with conventional electron microscopy⁵⁸ and also with cryo-CLEM³¹. Moreover, SVs in GLU synapses become elongated in the absence of VGLUT1, which reflects a lower luminal osmotic pressure^{59,60}. Therefore, DA vesicles, like GABA vesicles, may experience different osmotic pressure than GLU vesicles.” We now cite Nirenberg et al. 1996 (ref 57)

8. In their results section, the authors allude to the interesting possibility that the larger varicosities that also appear to form synapses are perhaps sites for glutamate release by dopamine neurons that express VGLUT2. This is certainly a possibility. The two papers that are cited to support this statement are not the most appropriate. The first reports of glutamate release by DA neurons and of VGLUT2 expression by these neurons are from the following two publications, which should be cited instead of the two that are presently cited. PMID: 15009640 PMID: 9614234.

We thank the reviewer for pointing this out. We have now added Dal Bo et al. 2004 (PMID: 15009640), to account for the first evidence of the presence of VGLUT2 in cultured DA neurons. We chose not to include Sulzer et al. 1998 (PMID: 9614234) because this earlier publication showed the possibility of glutamate co-transmission in DA neurons in culture, without reference to VGLUT2 (not identified at that time). Instead, we now cite also Stuber et al. 2010 (PMID: 20554874) which demonstrate with optogenetics and conditional VGLUT2 KO that this protein is involved in glutamate co-transmission in DA neurons prominently in the shell of the nucleus accumbens, a phenomenon much sparser in the dorsal striatum.

We cite these articles both in the Results section and in the Discussion. We have changed the sentence to: “They may correspond to terminals of DA/GLU neurons co-expressing the vesicular glutamate transporter VGLUT2 in vivo^{42,63,64} and also in cultured DA neurons⁶⁵. In

the adult mouse, these terminals are mostly concentrated in the shell of the nucleus accumbens⁶⁴.”

9. On page 11 of the results section, the authors state that “The percentage of tethered vesicles among proximal vesicles in DA is 25 ± 6 %, which was lower, but not significantly different from GLU synaptosomes (Figure 4C).”. The sentence should be modified to remove “which was lower”. If the difference is not significant, it is not appropriate to mention that it was lower. Same problem for a sentence in the section describing the results of figure 6: “The fraction of tethered vesicles was about 70 % bigger in GLU DA+ terminals but did not reach significance.”. The sentence needs to be modified accordingly.

With the new dataset of 24 GLU and 27 DA synaptosomes, the difference is now significant for the data presented in Figure 4C. For Figure 6H, we adjusted the sentence “The fractions of proximal vesicles that are tethered are not statistically different between GLU DA+ (61%) and GLU DA- (47%) (Figure 6H, p-value = 0.204)”

10. It is strange that in panel 4L, a t-test was performed to compare the proportions, but no error bars are shown. This needs to be clarified.

The reviewer is right. The graph of Figure 4L, is only a description of the proportions between the types of vesicles (tethered and/or connected, or not) in GLU and DA synaptosomes. We have removed the statistical test, which was not appropriate. In the text we just describe the proportions: “Finally, among proximal vesicles, 17% of GLU SVs are both tethered and connected vs 8.1% in DA synaptosomes. In the latter, the majority of vesicles are neither tethered nor connected (Figure 4L).”

11. The section of the results linked to figure 5 is the weakest of the study. The main conclusions are based on only 6 dopaminergic synaptosomes that contained multiple tethered vesicles. The conclusion that the location where most of the tethered vesicles is a putative active zone is perhaps correct, but there is simply not enough data to support it. Globally, this figure does not add much to the manuscript and could easily be removed.

To complement the previous dataset of 33 high resolution DA synaptosomes analysed for the presence of tethers (among which 13 had at least one tethered vesicle and 6 more than 2), we have acquired and analysed 16 additional tomograms of DA synaptosomes. Among these 16 new tomograms, 5 have at least one tethered vesicle and 3 more than 2. Therefore, the dataset presented in the revised version of the manuscript comprises 49 high resolution tomograms for which we could run the search of tethers and connectors. We found 18 synaptosomes with at least one tethered vesicle and 9 with at least two tethered vesicles. Therefore, the proportion of synaptosomes with tethered vesicles (18/49, 37%) is similar to the one we found previously (13/33, 39%). We have now a dataset of 45 tethered vesicles and 39 in synaptosomes with more than 2 tethered vesicles, so we think we have now enough data to conduct the analysis of the possible clustering of tethered vesicles.

Moreover, as requested by another reviewer, we have conducted two additional analyses. First, we have analyzed the clustering of tethered vesicles in GLU synaptosomes. We now show in Figure 5K that the minimal distance between pairs of tethered vesicles is significantly smaller in GLU terminal than in DA terminals, but that they are both lower than the distance between randomly placed tethered vesicles. Second, we have compared the size and sphericity of the tethered vesicles. We show that tethered vesicles are bigger and rounder than the general population of vesicles in DA synaptosomes (Figure 5F,G).

12. I don't think that it is appropriate to use multiple t-tests to compare the results shown in

panel 6K because the data sets are linked. The authors should reconsider this and consult a biostatistician.

We apologize for the wrong use of multiple t-tests in this situation. We now show only the percentages as for figure 4L, and draw the descriptive conclusion that proximal vesicles are more likely to be tethered and connected in DA + vs. DA- (27% vs. 5%).

13. In the discussion, the authors write “Therefore, we propose that synaptosomes containing tethered vesicles correspond to release-competent DA terminals.” This conclusion is not the only interpretation of such results. I invite the authors to consider other published results with higher sensitivity techniques suggesting that in fact, a majority of axonal release sites established by DA neurons are release competent, even if they do not express high levels of active zone proteins. See for example PMID: 34320240.

We now cite Ducrot et al. 2021 (PMID: 34320240) which showed that, similar to Liu et al. 2018 (ref 12), a subset of DA varicosities contains the AZ proteins bassoon, RIM or ELKS. We now tone down our conclusion: “Interestingly, other studies have shown that only ~30 % of DA axonal varicosities contain active zone proteins bassoon, RIM or ELKS^{12,14}, and this proportion is also found for functional DA varicosities able to release fluorescent dopamine analog¹⁵. Therefore, we propose that synaptosomes containing tethered vesicles correspond to DA terminals with highest probability of dopamine release.”

More minor issues:

1. On page 8, what the authors mean by «The normalized densities decrease by 20% in both compartments» is unclear.

We have rephrased this sentence: “We analysed the electron density in close proximity to the pre- and post-synaptic membranes of GLU synaptosomes. This may reflect the density of proteins and lipids in the area. The density is maximal at the vicinity of the plasma membrane. At both sides of the synapse, we measured a decrease by ~20 %, but within 10 nm distance away from the plasma membrane it in the pre-synapse while 40 nm are required to reach this percentage of decrease at the post-synapse”

2. In all bar graphs, the individual data points should be illustrated.

We have now added individual data points in all bar graphs.

Reviewer #2 (Remarks to the Author):

What are the noteworthy results?

This work provides a comprehensive structural characterization of synaptosomes isolated from mouse striatum. The novel and noteworthy results arise from the isolation of dopamine hub synaptosomes which contain dopaminergic terminals in apposition to glutamatergic synapses, and the characterization of all structures involved.

- Will the work be of significance to the field and related fields? How does it compare to the established literature? If the work is not original, please provide relevant references.

This work provides structural characterization that could support future studies investigating the regulation of striatal glutamatergic synapses by dopamine release. This manuscript provides more comprehensive characterization of “dopamine hub synapses” that have previously been isolated and characterized by the same group.

- Does the work support the conclusions and claims, or is additional evidence needed?

Given that the manuscript is primarily descriptive in nature, there is not a strong claim. The data provided does support the hypothesis that dopamine hub synapses (or synaptosomes) are unique structures, there are differences between glutamatergic and dopaminergic terminals involved in the DHS, and that there may be small ultrastructural differences in GLU synapses that are contacted by DA terminals compared to those that aren't.

- Are there any flaws in the data analysis, interpretation and conclusions? - Do these prohibit publication or require revision?

There are several minor revisions that must be made to the manuscript before publication. See the attached list.

- Is the methodology sound? Does the work meet the expected standards in your field?

Yes. The study has been well-designed and clearly communicated.

- Is there enough detail provided in the methods for the work to be reproduced?

Yes. The methods are comprehensive and well-explained.

We thank the reviewer for his/her positive evaluation of our manuscript. We have addressed all his/her specific points below.

Suggested edits

New analyses:

- From the main text: "First, the active zone could locate at or right next to the contact zone with GLU terminals. However, tethered vesicles are almost never found at the contact site." – quantify where the tethered vesicles are located relative to the contact site. For example, analyze distance along the plasma membrane from the center of contact site to the vesicle tether

We have now analyzed the distance of tethered vesicles from the center of the contact site (Figure 5H). We write: "However, tethered vesicles are almost never found at the contact site. While we found only one tethered vesicle at the DA/GLU contact site, the other 44 are distant from 250 to 750 nm. Indeed, the average distance of tethered vesicles from the contact site is 479.8 ± 292.2 nm (Figure 5H)."

- Comment on the sphericity of tethered DA vesicles compared to the average sphericity for all vesicles (or all untethered vesicles)

We have now quantified the sphericity and size of tethered vesicles and reported it in Figure 5G. We write: "We also compared the diameter and sphericity of the 43 tethered vesicles with the whole population and found that tethered vesicles are larger (Figure 5F, $p = 0.0016$) and rounder (Figure 5G, $p < 0.0001$)."

- Two different labelling methods were used to identify and characterize GLU and DA synapses (AAV vs genetically encoded). It seems that synaptosomes from both labelling methods were pooled for analysis of the characteristics of individual DA and GLU synapses. If not, please specify in figure captions and in main text which method was used. If so, please provide evidence that there is no significant difference in characteristics (size, vesicle size and occupancy) between synapses from AAV labelled and Ai14-tdtomato labelled samples.

Comparing DA vesicles from the two data set we found that the maximal diameter of vesicles from the two preparations is: 46.22 nm with AAVs and 48.56 nm with tdTomato; the sphericity is also comparable: 0.972 with AAVs and 0.979 with tdTomato. Additionally, the density of vesicles in the synaptosomes is not statistically different ($614 \text{ ves}/\mu\text{m}^3$ vs. $513 \text{ ves}/\mu\text{m}^3$). Both

groups are consistent and the phenotype is significantly different with GLU terminals for each labelling method.

- How often DA terminals appose a membrane that is not recognizable as a GLU terminal?

Most of the time DA are found apposed to membranes that are not recognizable as GLU terminals. Based on previous publications, they contact GABAergic (~30%) and Cholinergic (~15%) boutons which are abundant in the striatum. We employed cryo-CLEM in order to describe specifically the interaction between GLU and DA terminals that were also reported in the tissue (Pickel 1981; Nirenberg et al., 1996; Moss and Bolam, 2008).

- Quantify location of DA/GLU contact site compared to active zone – for example, analyze distance along the plasma membrane from the center of the active zone to the center of the contact site

We quantified the distance from the center of the active zone to the center of the contact site. We now report this distance in the Results section: “The distance between the centers of the GLU AZ and the DA/GLU contact site is 859 ± 389.4 nm.

- From text: “However, the absence of a morphologically identifiable DA active zone, may have biased the measurement of occupancy, because it included a larger, possibly irrelevant volume in our analysis.” This is an important point. One way to work around this bias is to perform similar analysis for GLU terminals. In other words, quantify the number of vesicles in GLU terminals that are proximal to the whole plasma membrane rather than just the active zone. Are there any instances of GLU SVs that are tethered outside the active zone? If so, how many?

Indeed, we find in about 65% (17/26) of GLU synaptosomes where tethers have been quantified, tethered vesicles out of the active zone. This phenomenon has been reported using cryoET (Fernández-Busnadiego et al., 2010).

- The minimum distance between tethered vesicles (along the plasma membrane) is quantified for DA synaptosomes. Please provide the minimum distance measurements for GLU synaptosomes also.

We have now quantified the minimum distance between tethered vesicles along the plasma membrane in GLU synaptosomes (Figure 5K)

Main text revisions:

- Please include a citation for the following statement “Importantly, CS-DHS contain an increased signal for the presynaptic proteins VGLUT1 and bassoon compared to other CS synapses”

We have added Paget-Blanc et al. 2022.

- From text: “The normalized densities decrease by 20 % in both compartments”
 - o When characterizing the electron densities/protein composition, it’s implied in the figure that this decrease is comparing cytoplasmic protein density to plasma membrane density but this is unclear in the main text. Please clarify this statement in the main text to what this decrease refers.

We changed the text to make it clearer: “We analysed the electron density in close proximity to the pre- and post-synaptic membranes of GLU synaptosomes. This may reflect the density of proteins and lipids in the area. The density is maximal at the vicinity of the plasma membrane. At both sides of the synapse, we measured a decrease by ~20 %, but within 10 nm distance away from the plasma membrane it in the pre-synapse while 40 nm are required to reach this percentage of decrease at the post-synapse (Figure 3J,K).”

- “Another DA synaptosome has 295 vesicles (see two upper dots in Figure 2E) and also resembles classical GLU synaptosomes, even though it was not connected to a PSE but engaged into a CS-DHS”
 - o This sentence implies that the two dots refer to the DA synaptosome with 295 vesicles and no PSE but based on the figure, these two dots correspond to the DA terminals that have a PSE (with ~600 and 40 vesicles). Please clarify in the main text, add all 3 outlier points to the bar charts, and explain what the dots correspond to in the figure caption.

We apologize for this misleading sentence, the two dots correspond to 2 DA synaptosomes with a PSE. These are the one comprising 610 and 39 vesicles.

General revisions to manuscript content

- “Overall, we did not identify a localized site for DA vesicle tethering, suggesting a large AZ for exocytosis” – I don’t think this is an appropriate claim because this could also mean no “active zone” especially given the lack of evidence of any dopamine release in these synaptosome preps (in other words, there’s no evidence that the DA synaptosomes specifically are active).

We have changed the last sentence of this paragraph for: “Overall, we did not identify a localized site for vesicle tethering, suggesting a different organization for vesicle tethering in DA and GLU synapses.”

- “Therefore, our data indicate that the contact between DA and GLU terminals at DHS modulates the SV organization at the single nanometer scale, as well as release properties of GLU terminals, and that this modulation is mediated by SV connectors.” – This is not an appropriate claim. There is no evidence of differential release properties of GLU terminals here – just slight structural changes.

We have removed “as well as release properties of GLU terminals”.

- Not sufficient evidence for this statement: “Moreover, in T+ DA synaptosomes, vesicles are located closer to the plasma membrane than in T- DA synaptosomes (Figure 5G), reinforcing the idea that in DA synaptosomes vesicle location is polarized to a putative active zone where vesicle may tether and fuse”

We have removed “reinforcing the idea that in DA synaptosomes vesicle location is polarized to a putative active zone where vesicle may tether and fuse”

- Therefore, we propose that synaptosomes containing tethered vesicles correspond to release-competent DA terminals – This is purely speculative as no data is provided that dopamine is actually released from these synaptosomes. Wished there was at the least Bassoon/RIM labelling to the DA synaptosomes, given that ~30% synapses that contain active zone proteins are the high Pr sites [PMC5807134].

We agree with the reviewer that this sentence is merely a prediction, although supported by work on GLU synapses. Nevertheless, the presence of bassoon in 30 % of axonal varicosities by Liu et al. 2018 is also correlative. There is no direct evidence that the varicosities with bassoon are the active ones.

Figure revisions

- Because this is primarily a descriptive publication, all figures and graphs would benefit from inclusion of individual data points rather than means along (relevant in figures...)

The figures and graphs were all revisited in order to include the data points when applicable.

- Figure 1:

o A, right, bottom: What does the black dotted line correspond to?

It corresponds to the striatum. The caption has been modified to mention it.

o Please indicate in both D (right) and E (left) where the insets (panel E, right) align
The D images the actual channels of the composite image showed on the right. In panel E we show an example of beads in both modalities (fluorescence and EM) that are present on the grid. We believe that showing where they are localized precisely is not necessary.

• Figure 2:

o C-F: as mentioned above, please explain in figure captions what the green dots correspond to

The dots corresponds to DA synaptosomes with a PSE, to explain it better, we added the following sentence in the caption of Figure C-F: "In the histograms of DA synaptosomes, orange dots represent the DA terminals with a PSE sharing the features of asymmetric synapses (examples in Figure S6)"

• Figure 3:

o A-C: quantify, either in a figure inset or in the figure caption, what proportion of segmented synaptosomes correspond to each DA/GLU/PSE configuration (2/94 for C, for example)

DA synaptosomes in DHS, contact either the GLU presynapse 28/32 (87%) or the GLU postsynapse 4/32 (13%), we now mention it in the text. However, illustrating the proportion of GLU/DA synaptosomes involved or not in a DHS (Figure 3A vs. 3B-C) would be confusing as we intentionally targeted the structures based on their fluorescent signal. The proportions do not represent the actual distribution in the brain. We detail this specific point in the discussion.

o E: Does the % of cleft distance start from the presynaptic or postsynaptic neuron? Explain x-axis in figure caption.

The percentage of the cleft distance Figure 3E is from the left to right of the interspace (between arrows), not from membranes. We apologize for this lack of clarity and adjusted the caption accordingly: "Pixel intensities along the length of the synaptic cleft (between the red arrows in D), normalized to neighboring background"

• Figure 4:

o In main text: "The percentage of inter-connected SVs was significantly higher in GLU (53%) than in DA synaptosomes (20%) (Figure 4J) and the length of connectors was significantly different (DA: 18.50 nm; std. 9.18 and GLU: 16.33 nm; std. 13.07; p-value = 0.0021) (Figure 4K).", text does not match figure.

We corrected these mistakes, now the text matches the figure panels.

o L: add error bars and individual points, as suggested above (what percentage per segmented synaptosome)

This figure shows the percentage of all vesicles tethered or connected (T+; C+). We apologize for the wrong use of error bars and statistical test. We adjusted the text accordingly, keeping it descriptive: "Finally, among proximal vesicles, 17% of GLU SVs are both tethered and connected vs 8.1% in DA synaptosomes. In the latter, the majority of vesicles are neither tethered nor connected (Figure 4L)."

• Figure 5:

o B,D, F: normalize to synaptosome volume (in addition or instead of raw numbers)

We normalize the number of vesicles to the volume (density) in a new graph Figure 5D. For clarity, we conserved the raw numbers of tethered vesicles and minimal distance to nearest neighbor, which the latest is now compared to GLU active zone.

o F: include a benchmark for the average distance to a random point on the membrane for comparison (as described in text)

We added a benchmark of the average random distance to a point on the membrane as described in the text.

- Figure 6

- o K: add error bars, would also be interesting to see the n for each bin (n would be helpful in all bar graphs)

Similar to the comment on Figure 4L above, we do not show the error bars because this graph plots the proportion of all vesicles. We added the corresponding n for each category. For all others bar graphs we added the n.

- Figure S1:

- o these are not violin plots – but would benefit from being violin plots (see above about including the individual data point distributions)

We now show the graphic Figure S1 as a violin plot.

- Figure S3/Table 2:

- o Expand table 2 to contain the “other” segmented organelles, for GLU terminals, DA terminals, and any organelles present in PSEs

We decided not to include the other segmented organelles because, they are either absent in most of the tomograms (e.g PSE often consists in a part of the membrane devoid of intracellular organelles) or cannot be identified with certainty (ER-like structures and endosome-like vesicles) because they are based on morphological features, naming them as such can be misleading. There are also other intracellular organelles of various shapes that did not fit in those two categories. Thus, we propose to keep the current Table 2 where only identified organelles are reported in order to avoid any misinterpretations.

- Figure S9:

- o G: add error bars

Error bars have been added to Figure S9.

- Fig S10:

- o A, C: Add individual values (per synaptosome segmented), error bars

- o A-D: Mention whether any of these comparisons are significant, either on figure or in figure caption

- o B: Function cumulates to 1 at only 20nm away from the plasma membrane, while proximal vesicles were defined as those within 45nm of the plasma membrane. Extend the x-axis to include all vesicles up to 45nm away from the membrane

The aim of this supplementary figure is to illustrate the distribution of tether lengths among the entire population of proximal synaptic vesicles. Tethers were binned in 6 nm intervals for clarity. Given the absence of known molecular significance for specific tether lengths in the current literature, we opted to keep this supplementary figure descriptive and did not perform statistical comparisons. However, in response to suggestions, we extended the x-axes in panels B and D to 45 nm to fully depict the observed range.

- Table 1: This is a somewhat confusing format. It would be more understandable if the singly labeled samples were listed in a separate table from the dually labeled samples, possibly include a venn diagram for each DA/GLU synaptosome and whether it was involved in a DHS, the “N” in the dually labeled synaptosomes presumably means that, for example, DA synaptosomes not involved in DHS appeared in 8/10 preps, but this is confusing as the total number of preps doesn't add up to 10. I don't find this particularly informative, especially given that there's no labeling of what each prep was (for example, there's no indication of what the overlap is for each prep)

We thank the reviewer for proposing a Venn diagram instead of the Table 1. However, we could not fit all the details as we do with the table. Thus, we reviewed the table 1 and included the new datas. There are indeed, 10 preparations for the bi-color strategy, among which there are 8 with DA not involved in DHS. With the 6 other preparations from mono-color mice, this makes it up to 16 preparations.

Reviewer #3 (Remarks to the Author):

Lapios et al. characterize the architecture of two major types of synapses found in dopaminergic neurons: glutamatergic and dopaminergic synapses. A key highlight of the study is the integration of cryo-electron tomography (cryo-ET) with fluorescence imaging to accurately depict striatal synaptosomes composed of dopaminergic (DA) nerve terminals. This work aligns well with previous findings. First, from serial section EM of DA neurons in situ, which initially revealed vesicle size pleomorphism (PMID: 34965204), and later, from a study suggesting that such pleomorphism is inherent to the type of vesicular transporters involved (PMID: 39788994). This was observed across multiple systems, including in vitro reconstitutions, iPSC-derived DA neurons, and striatal synaptosomes. Therefore, the publication of this manuscript is both timely and relevant to the emerging concept of non-canonical synapses, which likely represent the dominant mode of action in dopaminergic neurons. I am, thus, in favor of its publication, pending the authors' response to some reviewer comments.

1) In Figure 1G, what is the morphological difference between SV (blue) and endosome-like vesicle (yellow)? Is it possible to show the sizes of them? Do endosome-like vesicle increase when DA and Glu synapses contact?

The morphological difference between SV (blue) and Endosome-like vesicles (yellow) is the size. Vesicles smaller than 80 nm are considered synaptic, vesicles bigger than 80 nm are considered endosome-like vesicles. We quantified the size of endosomes in GLU and DA boutons and added a sentence in the Figure 2 results section: "Notably, although small (diameter < 80 nm) vesicles are clearly different between GLU and DA synapses, large endosome-like vesicles (diameter > 80 nm) have similar sizes (187 ± 122 nm $n = 63$ vs 179 ± 67 nm $n = 32$, for GLU and DA synaptosomes, respectively $p = 0.47$)."

2) In Figure 2G, the authors show the cumulative frequency distribution of vesicle diameter. Can the authors also include a dot plot to better depict the individual sizes of VGLUT+ vs DAT+ vesicles?

Here are the histograms for vesicle distributions for the three populations of synaptosomes (green for GLU, orange for DA with PSE, magenta for all DA):

Nevertheless, we think that the cumulative frequency distributions now shown in Figure 2H (new panel) are better to compare the distributions of vesicle size in the three types of synaptosomes.

3) The authors mention that 25% of the dopamine hub synapses (DHS) contact the cortico-striatal region. What about the remaining 75%, what do they connect to? Additionally, do dopaminergic neurons modulate only excitatory synapses, or do they also influence inhibitory ones?

This is a very interesting question as DHS appear to play a central role in the modulation of the striatal activity. We previously reported the actual proportion of synaptic association with other neuronal populations in the striatum (Paget-Blanc et al., 2022). DHS are also found with GABAergic boutons (27%) and cholinergic boutons (14%), as well as thalamo-striatal synapses (VGLUT2+, 12%).

4) I wonder if the size of the synaptosomes is also dependent on the presence or absence of mitochondria. Can a plot be shown for this feature?

We thank the reviewer for this comment. Indeed, there is a significant positive correlation between the size of the synaptosome and the presence of a mitochondrion, for both GLU and DA synaptosomes. We added this result in the text of the figure 2: “The synaptosomes with a mitochondrion are significantly larger than the ones without (max extension 1045 ± 205 nm vs 785 ± 250 nm, $p < 0.0001$.)” and “DA synaptosomes containing mitochondria (9/110) are significantly larger than the ones who do not (max extension 838 ± 224 nm vs 555 ± 176 nm, $p < 0.0001$.)”

5) How are inter-connected SVs measured, by physical distance or presence of connectors? Inter-connected SVs are measured based on the presence of a connector.

6) In Figure 5, do you think the distribution of more “flat” vesicles is dependent on their proximity to PM or completely independent of tethering properties?

Again, we thank the referee for this relevant question. DA tethered vesicles are significantly rounder compared to the whole population of vesicles. We added a graph corresponding to this observation in Figure 5G. Additionally, we also show that tethered vesicles are bigger (Figure 5F). Thanks to this comment, we believe that the observations described in the Figure 5 are reinforced.

7) Previous studies have shown that GABA (PMID: 23034651) can also be secreted from dopaminergic axons. Did the authors investigate VGAT-positive vesicles? Some GABA vesicles are known to have a more “flat” morphology.

Indeed, DA terminals can release GABA in acute striatal slice preparation. However, it was shown in this article by Tritsch et al. (2012) that GABA release was due to the presence of VMAT2, not VGAT, in DA terminals. Regarding the appearance of vesicles in GLU, DA and GABAergic terminals, we write in the Discussion: “Elongated-shaped vesicles have been documented in DA axons^{16,57} and at GABAergic inhibitory synapses with conventional electron microscopy⁵⁸ and also with cryo-CLEM³¹. Moreover, SVs in GLU synapses become elongated in the absence of VGLUT1, which reflects a lower luminal osmotic pressure^{59,60}. Therefore, DA vesicles, like GABA vesicles, may experience different osmotic pressure than GLU vesicles.”

8) If possible, it would be informative to assess the vesicle density on the plasma membrane during stimulation, particularly in dopamine varicosities. This could help determine whether the vesicles are functionally active or simply stored.

We agree with the reviewer that this experiment would be very informative. However, we do not have the capacity with our current equipment (Vitrobot Mark IV (Thermo Fisher Scientific)) to perform fast exchange of solution required for this experiment.

Reviewer #4 (Remarks to the Author):

Lapioš and colleagues present work describe a series of studies that employ a combination of cryo-correlative light and electron microscopy (cryo-CLEM) and cryo-electron tomography (cryo-ET) in the context of imaging mouse striatal dopaminergic and glutamatergic synaptosomal preparations. Using these approaches, the authors describe the three-dimensional architecture of glutamate and dopamine synapses. This includes providing novel details that distinguish these different families of synapses and their organization. Just as importantly, the manuscript employs ultrastructural reconstructions and thorough quantitation to define subpopulations of dopamine terminals according to vesicle tethering. Finally, this work analyzes differences between glutamatergic synapses that are connected to dopamine terminals versus those synapses that are not, offering structural details that further enhance the details structural characterizations of these synaptic structures.

Altogether, the above results presented in the manuscript are consistent with and build upon a robust earlier literature that has characterized both glutamatergic and dopaminergic synapses in mammals via room temperature transmission electron microscopy, serial volume electron microscopy, as well as as cryo-CLEM/cryo-ET to a lesser degree (e.g., PMID: 34965204, 29311144). Nevertheless, there are several key weaknesses that significantly diminish enthusiasm for the present work:

1) Foremost, the entire premise of the work is built around the use of purified synaptosomal preparations followed by fluorescence-based FACS sorting. Both of these methods are not gentle, but rather introduce substantial shearing forces to the structures being studied here - a fact that the authors acknowledge. Given the physical stresses introduced to the experimental system, and the fact that virtually all of the ultrastructural findings absolutely depend upon intact synaptic and vesicular structures, it is very difficult to rule out that the sample preparation may have distorted at least some of the reported findings. The ability to confirm the study's conclusions in a more native context would be strongly recommended. For example, the authors could conduct CLEM or serial EM (at room temperature or under cryogenic conditions) on intact cellular preparations examining these same dopaminergic and glutamatergic synapses.

We agree with the reviewer that the work we present in the manuscript is based on the preparation of synaptosomes from adult mouse striatum. We would like to point out that the work presented here does not use FASS, only the identification of GLU or DA synaptosomes by fluorescence without sorting. Moreover, as we write in the Discussion: “Synaptosomes obtained from mouse brain constitute a reliable model to investigate the spatial configuration of synapses^{17,38}. They are particularly amenable for cryo-ET because they can be directly observed by cryo-EM without further processing such as cryo-sectioning or cryo-focused ion beam milling¹⁷. Moreover, they retain functionality, such as SV exocytosis and endocytosis, protein composition and post-synaptic calcium signalling^{3,30,36,38}.”

Our study relies on the use of cryoET, not only to quantify the presence, characteristics (size and sphericity) and the position of vesicles, but also the presence of tethers, which can only be visualized unperturbed with cryoEM. This technique is achievable directly for synaptosomes or cultured neurons. We argue that studies on cultured DA neurons have shown so far the presence of large pools of small vesicles (Lycas et al. bioRxiv DOI 10.1101/2024.04.15.589543) which is the hallmark of immature DA neurons but not of DA neurons in adult mouse brain, as shown in Wildenberg et al. (2021 10.7554/eLife.71981) which we cite in the manuscript. Therefore, our results are the first to report the distribution of SVs and the presence of tethers in DA terminals from adult tissue. For this technique to be amenable for adult tissue, we would need to use cryo-sectioning or ion beam milling together with correlative light microscopy, techniques which are beyond what we can currently achieve in our laboratories.

2) Given the preponderance of heterogeneous vesicular structures in dopaminergic synapses, it is difficult to determine whether some of these structures are truly synaptic vesicles or other irregularly shaped organelles such as endosomes. This would be remedied by labeling of vesicles with a fluorescently-tagged vesicular monoamine transporter 2 (VMAT2) - a definitive marker of monoaminergic synaptic vesicles. Indeed, in one instance, the authors describe a dopamine synaptosome where most of the vesicles are small and round, which differs markedly from their other preparations, raising the possibility that this may not be a true dopaminergic synaptic structure. Having an additional tag like VMAT2 would increase the confidence in such a description.

In our manuscript, we use soluble fluorescent marker (mNeonGreen or tdTomato) to label DA synaptosomes and thus have an unbiased way to label all DA varicosities, even the ones mostly devoid of vesicles, as described in Wildenberg et al. 2021 (PMID: 34965204). Indeed, we find that 27/110 DA synaptosomes have less than 5 vesicles. This type of synaptosome labelled with VMAT2-Venus would most likely remain undetected. On the other hand, we agree that labelling VMAT2-Venus and co-labelling with e.g. tdTomato would be very interesting to decipher if part of the heterogeneity in vesicular content is due to different types of vesicles, as suggested by other publications (Zhang et al. Nature Neuro 2015 PMID: 25664911). Nevertheless, we think that this issue is beyond the scope of the present manuscript.

3) Finally, one of the defining features of dopaminergic synapses is the reliance on bulk neurotransmission which differs substantially from glutamatergic neurotransmission, particularly at type 1 synapses. Having a more detailed discussion of these differences would clarify and inform the Discussion section.

We discuss in the Introduction and the Discussion the issue of bulk versus synaptic transmission. However, we did not make the term explicit. We have added in the Introduction: “Like other neuromodulators, dopamine released from a single release site could influence large neuronal assemblies in a so-called volume transmission²⁰.”

Reviewer #5 (Remarks to the Author):

We appreciate the initiative by Nature Communication and the main reviewer to allow co-reviewing and acknowledge for it. It promotes training of junior colleagues and certainly improved the quality of the reviewing of our work.

Answer reviewers

Reviewer #1 (Remarks to the Author):

The revised manuscript corrects some of the issues raised in the first version. A few smaller issues still remain to be corrected:

1. Line 68: replace the word “terminal” by “varicosity” because a varicosity not containing any vesicles cannot really be called a terminal.

We agree with this point and replaced “terminal” by “varicosity”.

2. Lines 72-73: What the authors mean by “difficulties to identify rare DA axons” is unclear. DA axons in the striatum are not rare.

We have removed the word “rare”

3. Lines 208-209: The sentence beginning with “Notably” is incomplete and needs to be revised.

It was actually a typo. A dot should have been a comma, connecting this sentence with the next one: “Notably, although small (diameter < 80 nm) vesicles are clearly different between GLU and DA synapses, large endosome-like vesicles (diameter > 80 nm) have similar sizes (187 ± 122 nm n = 63 vs 179 ± 67 nm n = 32, for GLU and DA synaptosomes, respectively p = 0.47).”

4. As mentioned by the authors on lines 221-222, an active zone is typically defined, at least for glutamate and GABA synapses, as a membrane region in close contact with a postsynaptic membrane. Because the authors clearly show that this is very rare for DA neuron terminals, in keeping with much previous transmission EM work, do the authors conclude that DA neurons do not possess an active zone? In the discussion, the authors do refer to the idea that DA terminals appear to possess release sites that are different compared to glutamate terminals, but they do not make a very clear point about what they suggest. It would be good to clarify this.

For glutamatergic synapses, the definition of the active zone (AZ) is first morphological (the area of the presynaptic plasma membrane facing the post-synaptic membrane). Functionally, the AZ is also the part of the plasma membrane where SVs tether.

We have added in the Introduction (lines 49-51): “The AZ, in which SVs are thought to fuse with the plasma membrane to release neurotransmitter, contains specific proteins such as RIM1/2, bassoon and ELKS².”

For dopaminergic varicosities, we do not detect clear contact with a PSE, which would be a simple way to predict an AZ, i.e. the location where vesicle fusion occurs, presumably where AZ proteins are located. To make this clearer, we have added in the Results (lines 322-326): “In T+ DA synaptosomes we do not observe a clear AZ, which corresponds in GLU synapses to the location facing the synaptic cleft where tethered synaptic vesicles concentrate and undergo exocytosis upon stimulation. We looked for a putative DA AZ, which is a region of the plasma membrane where vesicle fusion may occur preferentially, in two ways.”

Finally, we added in the Discussion (lines 483-487): “Nevertheless, we found that in DA synaptosomes with multiple tethered vesicles, these tended to cluster, albeit with larger distances between vesicles than in GLU synaptosomes. This suggests that, even in the absence of a clear

morphological hallmark, molecular factors such as AZ proteins could direct vesicle tethering and fusion at specific sites in the DA varicosity.”

5. Lines 226-227: the sentence is unclear. What the authors are trying to say by “membrane it in the pre-synapse” is confusing.

We apologize for the confusion. We have changed the text to make it clearer: “We analysed the electron density in close proximity to the pre- and post-synaptic membranes of GLU synaptosomes (Figure 3J,K). This may reflect the density of proteins and lipids in the area. The density is maximal at the vicinity of the plasma membrane. At the presynapse, we measured a decrease by ~20 % in less than 10 nm away from the plasma membrane. In the PSE, we detected, in addition to the decrease within 10 nm, a clear increase in density 10 to 25 nm from the PSE membrane, which corresponds to the post-synaptic density (PSD) as detected with other modalities of electron microscopy in situ ⁴¹ and synaptosomes with cryo-ET ³¹.”